# Eighty Years of Data Collected for the Determination of Rainfall Threshold Triggering Shallow Landslides and Mud-Debris Flows in the Alps

Fabio Luino [1] , Jerome De Graff [2] , Anna Roccati [1,*], Marcella Biddoccu [1],
Chiara Giorgia Cirio [1], Francesco Faccini [1,3,*] and Laura Turconi [1]

[1]  Istituto di Ricerca per la Protezione Idrogeologica, Consiglio Nazionale delle Ricerche, Strada delle Cacce 73,
    10135 Torino, Italy; fabio.luino@irpi.cnr.it (F.L.); marcella.biddoccu@ima.to.cnr.it (M.B.);
    cgcirio@gmail.com (C.G.C.); laura.turconi@irpi.cnr.it (L.T.)

[2]  Department of Earth & Environmental Science, California State University, M/S ST24, Fresno, CA 93740,
    USA; jdegraff@csufresno.edu

[3]  Dipartimento di Scienze della Terra, dell'Ambiente e della Vita, Università di Genova, corso Europa 26,
    16132 Genova, Italy

*   Correspondence: anna.roccati@irpi.cnr.it (A.R.); faccini@unige.it (F.F.); Tel.: +39-010-3538039 (F.F.)

**Abstract:** Identifying the minimum rainfall thresholds necessary for landslides triggering is essential to landslide risk assessment. The Italian Alps have always been affected by shallow landslides and mud-debris flows, which caused considerable damage to property and, sometimes, casualties. We analysed information provided from different sources carrying on the most thorough research conducted for this alpine area. Thousands of documents and reports of rainfall values recorded over 80 years by rain gauges distributed in Sondrio and Brescia Provinces define the mean annual precipitation (MAP)-normalized intensity–duration thresholds for the initiation of shallow landslides and mud-debris flows. The established curves are generally lower compared to those proposed in literature for similar mountain areas in Italy and worldwide. Furthermore, we found that landslides occurred primarily at the same time or within 3 h from the maximum peak of rainfall intensity in summer events and in a period from 0 to 5 h or later in spring-autumn events. The paper provides a further contribution to the knowledge framework on the rainfall conditions required for the initiation of surficial landslides and mud-debris flows and their expected timing of occurrence. This knowledge is crucial to develop better warning strategies to mitigate geo-hydrological risk and reduce the socio-economic damage.

**Keywords:** rainfall thresholds; soil slip; mud-debris flows; Alps; Lombardy region

## 1. Introduction

According to the European Environment Agency (EEA) report on the impacts of natural hazards in Europe [1], almost 70 major landslides were recorded over the period 1998–2009 in different mountain areas, mainly in the Alps, the Scandinavian Peninsula, and the southern Europe. Collectively, these events claimed a total of 312 lives and caused economic losses and extensive damage to buildings and infrastructures. But the large landslides are not the only natural instability processes that cause problems to populations in the Alps. In fact, every year, several shallow landslides and mud-debris flows triggered by intense rainfalls occurred in many areas in the Italian Alps and similar mountain regions worldwide [1–14]. In these countries, they represent one of the most significant cause of death, damage, and destruction to property.

A large part of the public opinion and the scientific community suggest an increase in the number of rainfall events and, in turn, the future frequency and intensity of landslides is an expected impact of anthropogenic-induced climate changes [15–26]. However, it is difficult to make a clear long-term forecast of the increase (or decrease) of landslide risk under a changing climate because observed individual extreme events do not necessarily establish the frequency and intensity of landslides are statistical increasing. Additionally, the knowledge of the climatic variable that may modify the response of slopes to climate changes, the factors that control the initiation of failures, and the downscaling of climate modelling to a local scale required major research [1,25,27–32]. Analysis of historical data do not clearly show evidence of an increase trend in frequency with which damaging rainfall-induced landslides occur. Complicating understanding the role of climate change is the impact of geo-hydrological processes often being worsened in recent years by irresponsible land-use management, including uncontrolled urban sprawl and soil sealing, particularly in mountain areas.

Over the past decade, the improvement of knowledge and understanding the mechanism of natural instability processes, e.g., shallow landslides and mud-debris flows, have favoured more adequate and incisive measures in terms of forecast, prevention, and mitigation of mass movement risk directed to a better urban management resulting in reduced impacts, especially the loss of human lives and assets. Furthermore, the technological advances have enabled development of always more efficient real-time monitoring systems, useful to assist the planners and decision-makers of local administrations in planning and managing measures of civil defence in case of emergency [33–39].

The mountain territory of the Lombardy region, in Northern Italy (Figure 1), is historically affected by numerous natural instability processes triggered by heavy rainfalls. The Lombardy alpine and pre-alpine districts are among the mountain areas in the Alps and in the Italian and European regions most prone to landslide and with the largest number of properties and people exposed to a high and very high landslide risk [1,40–42] Shallow landslides and mud-debris flows frequently occurred, causing severe damage to structures and infrastructures and sometimes casualties, with considerable losses in socio-economic terms [6,43–45]. Shallow landslides initiate mainly on steep and very steep slopes characterized by eluvial and colluvial deposits and involve small thickness materials [46,47]. They are triggered by short but very intense rainfalls in summer or prolonged precipitations in spring and autumn. Despite the small volumes involved, they frequently caused extensive damage to man-made structures due to their widespread spatial distribution across territories, rapid development, and high velocity of propagation toward the valley floor. They can be particularly destructive when channelled in narrow and steep incisions, characterized by the presence of great amount of debris, thereby evolving into mud-debris flows. The solid-liquid mixture of a mud-debris flows is composed by water, fine materials, pebbles, and large-size boulders accompanied by shrubs and uprooted trees. The moving materials flow like a rapid to very rapid fluid, and flowing downstream, they increase their volume and their destructive power [47–49].

In recent decades, the regional administration financed several research programs aimed to forecast and prevent the natural instability processes and to mitigate their potentially destructive impacts on structures, infrastructures, and on population [50,51]. The knowledge of the rainfall conditions required to trigger shallow landslides and mud-debris flows is essential for forecasting rainfall-induced mass-movements and the implementation of an efficient operational early-warning system for civil defence purposes [34,52–56]). However, it is often difficult to issue a warning with a specific alert time because of to the lack of information for a real-time assessment of these dangerous natural processes and their very to extremely rapid development.

In this paper, we discussed the rainfall events that have triggered shallow landslides and mud-debris flows in the alpine and pre-alpine catchments in the Lombardy region and the Central- and South-Eastern Alps [57] during the period 1927–2008. The systematic analysis of the historical rainfall and landslide information identified 661 events during that period. Using a subset of 291 events for which location and timing of landslides occurrence were exactly known and validated hourly rain measurements were available, we defined rainfall thresholds for the possible initiation of shallow

landslides and mud-debris flows in the alpine provinces of the Lombardy region. We compared the new thresholds with similar published curves for alpine and mountain areas in Italy and worldwide and highlighted their useful implementation in the regional landslides warning model in order to increase the system efficacy and assess the potential hazardous scenarios related to shallow landslides and mud-debris flows along slopes and alluvial fans in the alpine catchments at regional scale.

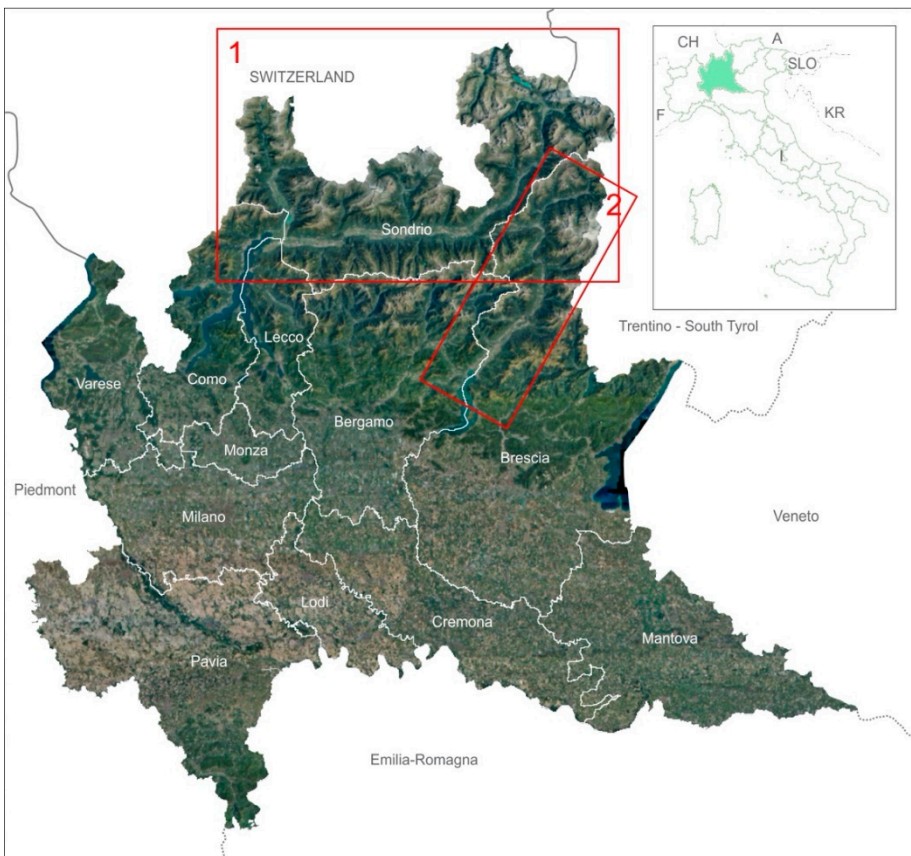

**Figure 1.** Location of the study areas in the alpine sector of the Lombardy Region: (**1**) Sondrio Province (Valtellina); (**2**) Brescia Province (Valcamonica).

Furthermore, we provided a further contribution to the knowledge framework for the development of real-time assessment of mud-debris flows in different mountain catchment areas in order to plan the most effective mass movement risk prevention and mitigation strategies.

## 2. Past Studies and Methodologies

In recent decades, several studies have investigated the role of rainfall in triggering instability processes worldwide and highlighted the cause-and-effect relationship between the characteristics of the pluviometric events and the initiation of mass movements, regardless of the morphological settings of the study area. In literature, different methods have been proposed to analyse rainfall-induced landslides. Physically-based approaches rely on the understanding and the evaluation of the complex physical and hydrological models that influence the stability condition of the slope [58–65]. They require numerous parameters that differ greatly in space and time, particularly across large regions (e.g., changes in rainfall regime, effects of vegetation, and mechanic and hydraulic properties of both soils and bedrock). Consequently, physically-based methods are generally too elaborate to apply and perform on definite portion of the slope or a single catchment. Empirical approaches count on the statistical analysis of the rainfall conditions that triggered landslides in the past in order to define rainfall thresholds for the possible initiation of landslides and mass movements [12,34,52,66–72]. They

require a large catalogue of both landslide and rainfall events and a careful evaluation due to the wide variability of the various physical, geological, geomorphological, and climatic factors that influence the initiation of landslides.

Empirical approaches proposed by authors differ in: (i) type of mass movement (e.g., shallow landslides, debris flows, etc.), (ii) rainfall data (e.g., intensity and duration of rainfall event, mean annual precipitation, cumulated event rainfall, antecedent rainfall, etc.), and (iii) geographical validity of the rainfall threshold (e.g., global, regional, local, etc.). In the following paragraphs, the most important works aimed to define the rainfall thresholds for the initiation of shallow landslides and mud-debris flows, particularly in the Italian Alps and secondarily in similar mountain environment in Italy and different regions worldwide, are briefly described.

Caine [66] proposed a global intensity–duration threshold for triggering shallow landslides and debris flows. It was obtained by plotting the lower curve that fitted 73 rainfall conditions correlated to landslide occurrence worldwide in different geological, morphological, and climatic conditions for which temporal and spatial information were known. Govi and Sorzana [73] analysed precipitations that triggered several shallow landslides in some alpine catchment in the Piedmont region and defined a rainfall threshold for mud-debris flows based on the mean annual precipitation (MAP). Initiation of the mass movements was strongly influenced by the rainfall intensity and the antecedent cumulative precipitation, in addition to the type of soils and the slope acclivity. Later, Govi et al. [74] observed that hourly rainfall intensity and cumulative rainfall (correlated to the MAP) influence the timing of occurrence of landslides and defined two different curves for events that take place in winter-spring and summer-autumn months, respectively. Analysing 22 rainfall events that partially triggered debris flows in Santa Cruz Mountains, Wieczorek [75] observed that antecedent rainfalls were decisive in triggering mass movements and defined a new rainfall intensity–duration threshold, lower than the curve proposed by Caine. In order to make comparable rainfall thresholds established for different regions, Cannon [76] and Jibson [77] included the empirical rainfall measures of the local climate: they defined normalized ID rainfall thresholds for the possible initiation of debris flows and mud-debris flow in some regions worldwide, dividing the event rainfall intensity by the corresponding mean annual precipitation (MAP). In detail, Cannon [76] defined a threshold line by comparing the values of the normalized rainfall parameters of six storms that have accompanied abundant debris flows activity in the study area. The threshold was constructed by drawing a line that separated the rainfall conditions unique for an event that resulted in debris flows from more commonly occurring storm-rainfall conditions. In 1992, Ceriani et al. [78] applied the curve obtained by Govi et al. [74] for the Central- and South-Eastern Alps to the geo-hydrological events of July 1987. Analysis of the intensity-cumulative rainfall diagram for the observed events revealed that such seasonal curves could not be applied to that study area. For this reason, the authors defined a new threshold for the initiation of debris flows in the Lombardy Region using a catalogue of rainfall data recorded by rain gauges located in the Valtellina valley and information on instability processes, including debris flows, debris torrent, and shallow landslides, occurring within a 5 km buffer from the rain gauges over a 70-year period. Afterward, they improved the threshold, taking into account the rainfall events that also affected the Central- and South-Eastern Alps in summer 1992 and autumn 1993, in order to obtain two new different curves based on the mean rainfall intensity and the MAP-normalized rainfall intensity, respectively. The two obtained curves have been used to develop the regional early-warning system to forecast the occurrence of debris flows and shallow landslides. Thresholds for 12-, 24-, and 48-h periods have been estimated so that mass movements may occur for rainfall conditions lower in only 5% of each alert homogeneous area. Bottino et al. [79] analysed the instability processes that affected the Ivrea sector (close the famous Morainic Amphitheatre) in November 1994, including slides and rock falls which developed into mud-debris flows. Using rainfall and landslide information over a 30-year period in the study area, they defined two curves that differ in the antecedent cumulated event rainfall for a period of 30 days before. Paronuzzi et al., [80] defined a MAP-normalized ID threshold for the North-Eastern Alps, by analysing rainfall data of 12 rainfall events that induced

debris flows and adopting the lowest curve as a minimum threshold. Similarly, Aleotti et al. [81] determined the local thresholds in some alpine valleys in Piedmont by analysing rainfall conditions that triggered soil slips in this area in October 2000. Wieczoreck [82] examined the rainfall intensity and duration characteristics of 27 June 1995, and other storms in the Blue Ridge of central Virginia and defined a minimum threshold necessary to trigger debris flows in granitic rocks. Later, Bertolo and Bottino [83] proposed a new approach to define rainfall thresholds for different western alpine valleys based on the analysis of heavy rainfall events from 1950 to 2000 that triggered different types of mass movements, including debris flows, mud-debris flows, and shallow landslides. For each alpine valley, they defined two curves: they correlate (i) the daily maximum rainfall value to the cumulative rainfall event, and (ii) the maximum daily precipitation recorded during the rainfall event and the cumulated rainfall that included the antecedent precipitation for a period of 15 days, with reference to the seasonal distribution. However, the approach proposed by the authors is reliable for local analysis, according to Bacchini and Zannoni [84], who analysed five rainfall events that triggered mud-debris flows in the Cancia alpine catchment, in the Dolomiti Mountains, frequently affected by rainfall-induced landslides. They observed that rainfall thresholds defined in terms of mean intensity, duration, and mean annual precipitation (MAP-normalization) were suitable at a regional scale but inadequate for a large-scale analysis. Furthermore, the curves were lower than those defined by Govi [73,74] and Ceriani et al. [67] because they were influenced by the geographical extent and the local climate conditions of the basin or the mechanical features of the mass movements. Aleotti [34] collected rainfall and landslide information on a large sector of the Piedmont Region affected by intense rainfall events over a 15-year period. First, the author investigated the antecedent rainfall for periods of 7, 10, and 15 days before the rainfall events and cumulated rainfall of the triggering day, but no correlation has been found. Later, he established the intensity–duration curve that leaves 90% of the points representative of rainfall conditions in the portion of the chart where landslides were expected. The obtained threshold was comparable to the curves defined by Caine [66] and Ceriani et al. [67] for some worldwide regions and the Lombardy Region, respectively. Furthermore, Aleotti highlighted the importance of the detailed compilation of a rainfall and landslide catalogue that include: (1) the accurate identification of the rainfall start-time, (2) the triggering time of the slope failure, (3) the area affected by the instability phenomena, and (4) the most representative gauge for the rainfall event. Giannecchini [85–87] analysed several events that triggered (or did not trigger) instability processes in the Apuane Alps and the neighbouring Serchio River catchment, Northern Italy: using a manual fitting methods by determining a differentiation line of the dataset, for each rain gauge in the study areas, the author defined two different intensity–duration thresholds for the possible initiation of shallow landslides and debris flows based on different values of MAP and duration of the rainfall event. The two curves divided the charts in three different fields corresponding to three different rainfall event categories: (1) events that triggered several damaging shallow landslides, (2) events that triggered a small number of shallow landslides, and (3) events that did not triggered landslides or for which information about rainfall-induced slope failures were not available. In terms of stability, the two threshold curves could reasonability delimit three fields of stability: certain stability (under the lower curve), uncertain stability (between the two curves), and instability (above the upper curve).

Brunetti et al. [88] proposed two statistical approaches for the definition of objective and reproducible rainfall thresholds for the Abruzzo Region (Central Italy), including a Bayesian inference method, formerly applied by Guzzetti et al. to define the minimum intensity–duration *ID* and normalized-*ID* thresholds for the initiation of landslides in the central and southern Europa [89] and worldwide [69]; and a new method based on a Frequentist probabilistic approach that best performs with large datasets. The latter method allows determining multiple curves for different exceedance probability levels, and it is functional for the development of probabilistic schemes for predicting possible landslide occurrence based on rainfall measurements or forecasts that can be implemented in landslide warning systems operating at different geographical scales. Both the empirical methods proposed by Brunetti et al. [88] have been adopted to define rainfall thresholds for the landslide

prediction in some districts of the Northern Apennine. The Bayesian statistical approach has been used by Cevasco et al. [90] to determine the rainfall thresholds for triggering shallow landslides on the Genoa municipality area: the two established curves separate the stability, intermediate, and instability fields in the *ID* bi-logarithmic diagram, but criteria used for drawing them are not clearly specified. Adopting a frequentist approach, a co-author in [72] recently defined the 5% *ID* threshold for the possible initiation of shallow landslides and hyper-concentrated flows in the Entella River basin, i.e., the threshold that leaves 5% of the empirical points that represent the rainfall conditions that triggered landslides below the established power law curve.

## 3. Study Area

The study area comprises two of the large and most important mountain areas of the Lombardy Region and the Central- and South-Eastern Alps (Figure 1): the Southern and Western Rhaetian Alps and Central Lombard Prealps, in the Sondrio Province, and the Central and Eastern Lombard Prealps, in the Brescia Province [57] (Figure 2).

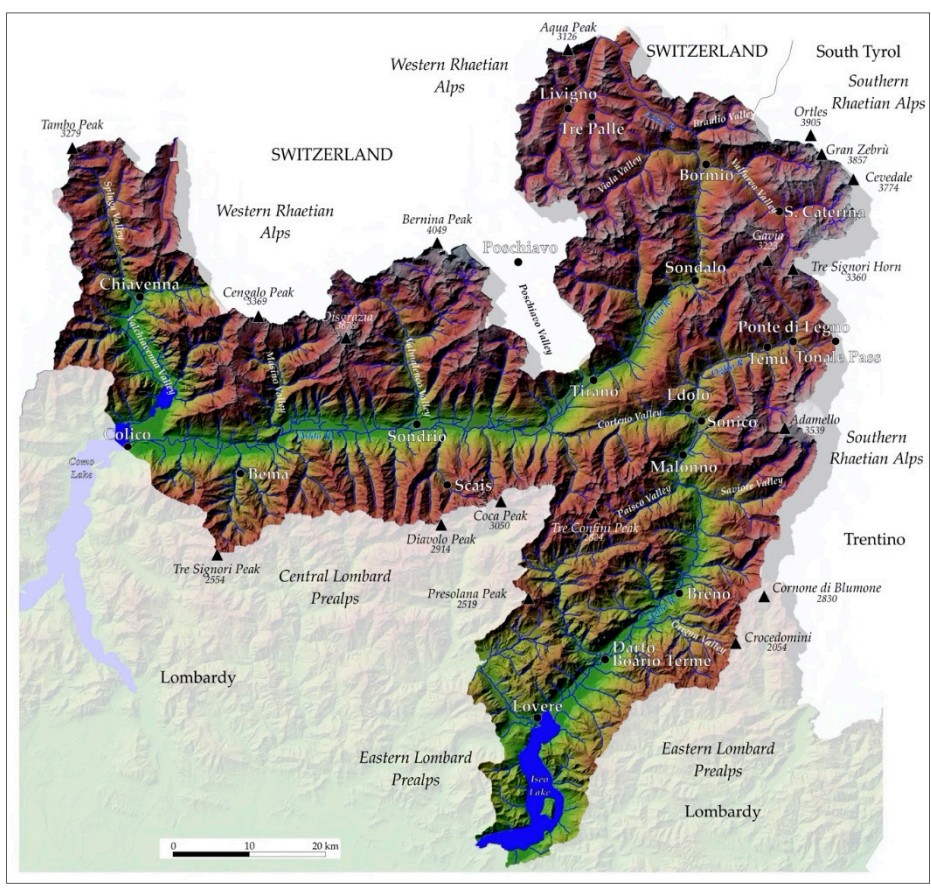

**Figure 2.** Geographical setting of the Valtellina (Sondrio Province) and Valcamonica (Brescia Province) in the Southern and Western Rhaetian Alps and the Central and Eastern Lombard Prealps (Central- and South-Eastern Alps) [45].

The Sondrio Province is occupied entirely by a mountain environment and includes the Valtellina valley and, secondarily, the Valchiavenna valley and the Livigno district. The Valtellina valley is a typical U-shape alpine valley derived from the Quaternary glacial processes and extends for 120 km in length between the Southern and Western Rhaetian Alps and the Central Lombard Prealps (Figure 2). It is enclosed by the Bernina Alps (Cengalo Peak, 3369 m; Bernina Peak, 4049 m; Aqua Peak, 3126 m) to the north, along the Swiss border; the Ortles-Cevedale Alps (Ortles, 3905 m; Gran Zebru', 3857 m; Cevedale, 3774 m) to the north-east, and the Orobian Alps (Tre Signori Peak, 2554

m; Diavolo Peak, 2914 m; Coca Peak, 3050 m; Tre Confini Peak, 2588 m) to the south. The valley has prevalently E-W direction resulting from its particular tectonic setting: it is superimposed on the regional fault "Insubric Line" that separates the Austro-Alpine nappe system to the north from the Variscan basement of the Southern Alps to the south. In the middle and upper sectors, the valley turns toward NE and N, respectively. Tectonics influence also the geological setting of the Valtellina valley: the bedrock consists mainly of metamorphic and intrusive lithotypes of the Austro-Alpine basement, included cataclastic and mylonitic rocks that crop up in the proximity of the tectonic lineament, with subordinate sedimentary rocks. Glacial, fluvio-glacial, and colluvial deposits of variable thickness cover the middle-lower portion of the slopes; thick alluvial deposits occupy the main valley floor, with several alluvial fans at the outlet of the tributary valleys. The Valtellina and its minor valleys, e.g., Masino, Valmalenco, Poschiavo, Valfurva, Braulio, and Viola valleys, form the upper Adda River basin from its sources in the Ortles mountains to its inlet into Lake Como.

Due to its peculiar orographic settings, insolation is very different along the slopes of the valley, particularly in autumn and winter months: the Rhaetian alpine slopes, facing south, get larger amounts of sunlight than the Lombard pre-alpine ones, facing north, with higher temperatures at the same elevations. The climate conditions of the Valtellina valley are largely continental, according to its terrain features and the orographic configuration of this alpine sector, with cold winter and hot summer. Generally, the coldest month is January, with mean temperature values ranging from −2.3 °C at Poschiavo, in the Bernina Alps area, to −0.8 °C at Bema, along the Central Lombard Prealps slopes, in the Lower Valtellina, and the warmest one is July, with mean values ranging from 15.4 °C at Poschiavo, to 22.4 °C at Sondrio, in the valley floor, in the Middle Valtellina [91]. The mean annual rainfall is about 1200 mm, with significant spatial variability due to the complex local orographic configuration (Figure 3): it ranges from 726 mm/y at Tirano, in the middle valley floor, to 1715 mm/y at Scais, in the Lombard Alps, and from 646 mm/y at Tre Palle, in the Livigno district, to 1822 mm/y in the Valchiavenna [92]. The MAP values gradually increase toward the Lario region, with three different regions: (i) a low MAP region (<1100 mm/y), in the Valmalenco valley, in the Medium and Upper Valtellina and in the Livigno district, (ii) a medium MAP region (1100–1300 mm/y), in the eastern Valchiavenna, in the Valmasino valley and in the Bernina sector, and (iii) a high MAP region (>1300 mm/y) in the Lombard Prealps sector of the Lower and Medium Valtellina and the western Valchiavenna. Precipitation is most abundant in autumn months, from October to November, in the Middle and Lower Valtellina and in summer months, from July to August, in the Upper Valtellina.

The mountain sector of the Brescia Province is represented by the Valcamonica valley, an alpine U-shape valley derived from the Quaternary glacial processes, that extends for about 90 km in the Central and Eastern Lombard Prealps, from the Cevedale and Adamello Alps to the Lake Iseo (Figure 2). It is enclosed by the Rhaetian Alps of the Upper Valtellina (Gavia Mount, 3223 m a.s.l.; Corno dei Tre Signori, 3360 m) to the north, the Adamello-Presanella Alps (Adamello Mount, 3539 m; Cornone di Blumone, 2830 m) and the Brescian Prealps (Crocedomini Mount, 2054 m) to the east; the Orobian Prealps (Presolana Peak, 2519 m) to the west and the Lake Iseo to the south. The valley is N-S oriented, except in the upper sector, where it has E-W orientation resulting by the Insubrian Line. Tectonics also influence the geological setting of the Valcamonica valley: in the upper and middle sectors, the bedrock consists mainly of intrusive rocks and metamorphic lithotypes of the Austro-Alpine and Southern Alps crystalline basements, whereas in the lower valley and in the Lake Iseo region crop up the Permo-Mesozoic and Paleocene sedimentary cover of the Southern Alps, including thick carbonatic and terrigenous successions, such as the typical sedimentary formations of the Lombardian Basin. Glacial, fluvio-glacial, and colluvial deposits of variable thickness cover the middle-lower portion of the slopes, whereas alluvial deposits occupy the main valley floor, with several alluvial fans at the outlet of the tributary valleys. The Valcamonica and its minor valleys, e.g., Corteno, Paisco, Saviore, and Grigna, coincide with the upper Oglio River basin from Ponte di Legno southwards to its inlet into Lake Iseo. Due to its local orographic configuration, the climate setting of the Valcamonica valley has a significant spatial variability. In the Lake Iseo region and in the Lower valley, NW-SE oriented and

enclosed by mountains moderate in elevation (lower than 2300 m a.s.l.), climate is typical Insubric, with a rainfall regime characterized by two peaks of precipitation in autumn (from October to November) and spring (from April to May) and higher temperatures due to the great insolation, except in the lake region where they are mitigated by the daytime breezes in spring and summer months. In the Middle Valcamonica, the N-S orientation of the valley and a relief that exceeds 2800 m in elevation reduce the insolation in the valley floor and the effects of the cold air masses from NE: consequently, strong west winds frequently descend, and disturbances from SW cause heavy rainfall. Thunderstorms commonly occur due to high relief, especially in summer months. Although temperature gradually decreases with the elevation, inversions are locally observed due to the morphology: the narrowing of the valley downstream Malonno facilitate the packaging of cold air masses into the flat land, causing an abrupt decrease of the minimum temperature values compared to Edolo, located at a higher altitude [93]. In the Upper Valcamonica, E-W orientation and high relief favour a typically continental climate: cold winter with snowfalls and hot summer with frequent thunderstorms, locally intense, with a maximum rainfall in autumn months (from October to November). Generally, temperatures gradually decrease northward and from low to high altitude. The coldest month is December, with mean temperature values ranging from 0.8 °C at Edolo, in the alpine sector, to 2.6 °C at Breno, in the Lower Valcamonica, and the warmest one is July, with mean values ranging from 19.7 °C at Edolo, to 23.2 °C at Breno [91]. Mean annual rainfall gradually decreases from Lake Iseo (1191 mm/y at Lovere) to the head of the valley (1085 mm/y at Sonico, 982 mm/y at Temu', and 1118 mm/y at Tonale Pass) (Figure 3) [92].

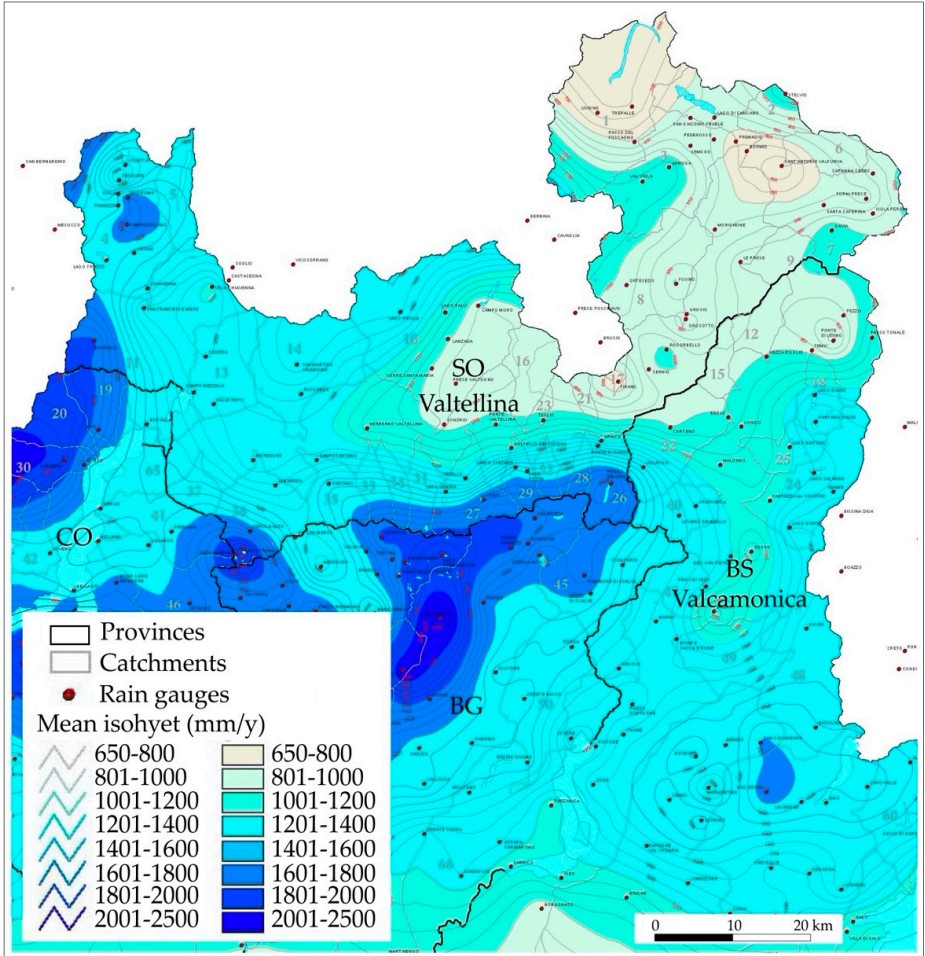

**Figure 3.** Map of the mean annual precipitation (MAP) in the 1891–1990 period [92] in the alpine region of Lombardy: BG, Bergamo Province; BS, Brescia Province (Valcamonica); CO, Como Province; SO, Sondrio Province (Valtellina).

Both in the Valtellina and Valcamonica valleys, the maximum number of rainy days is generally recorded in May (15 days on average) and the minimum in February (six days on average). However, according to the Bagnouls and Gaussen climate index [94], no dry period occurred: consequently, Adda and Oglio riverbeds are never totally dry. Minimum hydrometric levels are observed in the summer months or during prolonged dry spells, whereas the highest flows can be recorded during prolonged or very heavy rainfall events and late in spring and/or early in summer when snowpack melt and release water into alpine watercourses.

## 4. Materials and Methods

To define the rainfall conditions that may have triggered shallow landslides and mud-debris flows in the study area, we adopted a methodology structured in the following stages: (i) bibliographic research of historical information of rainfall-induced landslides occurred in the last century in the Lombardy Region, with particular reference to mud-debris flows and shallow landslides, in order to compile a regional georeferenced database of instability phenomena correlated to rainfall events primarily for the Sondrio and Brescia Provinces, (ii) research, analysis, and validation of pluviometrical data, (iii) definition of thresholds for the possible initiation of mud-debris flows and shallow landslides in the Rhaetian Alps and Lombard Prealps, and (iv) comparison to other curves proposed in literature for similar geographical regions.

### 4.1. Catalogue of the Historical Landslides in the Lombardy Region

We obtained information of rainfall-induced mud-debris flows and shallow landslides which occurred in the territory of the Lombardy Region over the 80-year period 1927–2008 from different sources, including scientific papers, technical and event reports, historical archives of local municipalities and research institutes and public agencies working on geo-hydrological risk reduction, newspaper articles, and interviews with local inhabitants [95,96] (Figure 4). In particular, most of the information on instability processes have been gathered from the historical archive and newspaper and periodical library of IRPI-CNR located in Turin, which keeps the largest collection of documents, maps, photographs, and newspaper articles related to geo-hydrological processes that affected Northern Italy since 1800. We considered all scientific and technical papers that contain information about one or more rainfall-induced mud-debris flow and/or shallow landslide event at least in Lombardy, with particular attention to the built-up areas located at alluvial fans in the Sondrio and Brescia Provinces (Figure 5). Furthermore, historical analysis included unpublished papers and technical or event reports from regional archives of the Territory and Urban Planning Department of the Lombardy Region Administration, provincial and regional archives of Corps of Forest Rangers in Sondrio, Brescia, and Milan, and several archives of local municipalities historically affected by severe mud-debris flows and landslides in Valtellina and Valcamonica. We also obtained accurate landslide information from technical reports and catalogues compiled by the Regional Geological Monitoring Centre in Sondrio and old chronicles and regional and local newspaper and periodical articles in the municipal libraries of Sondrio and Brescia.

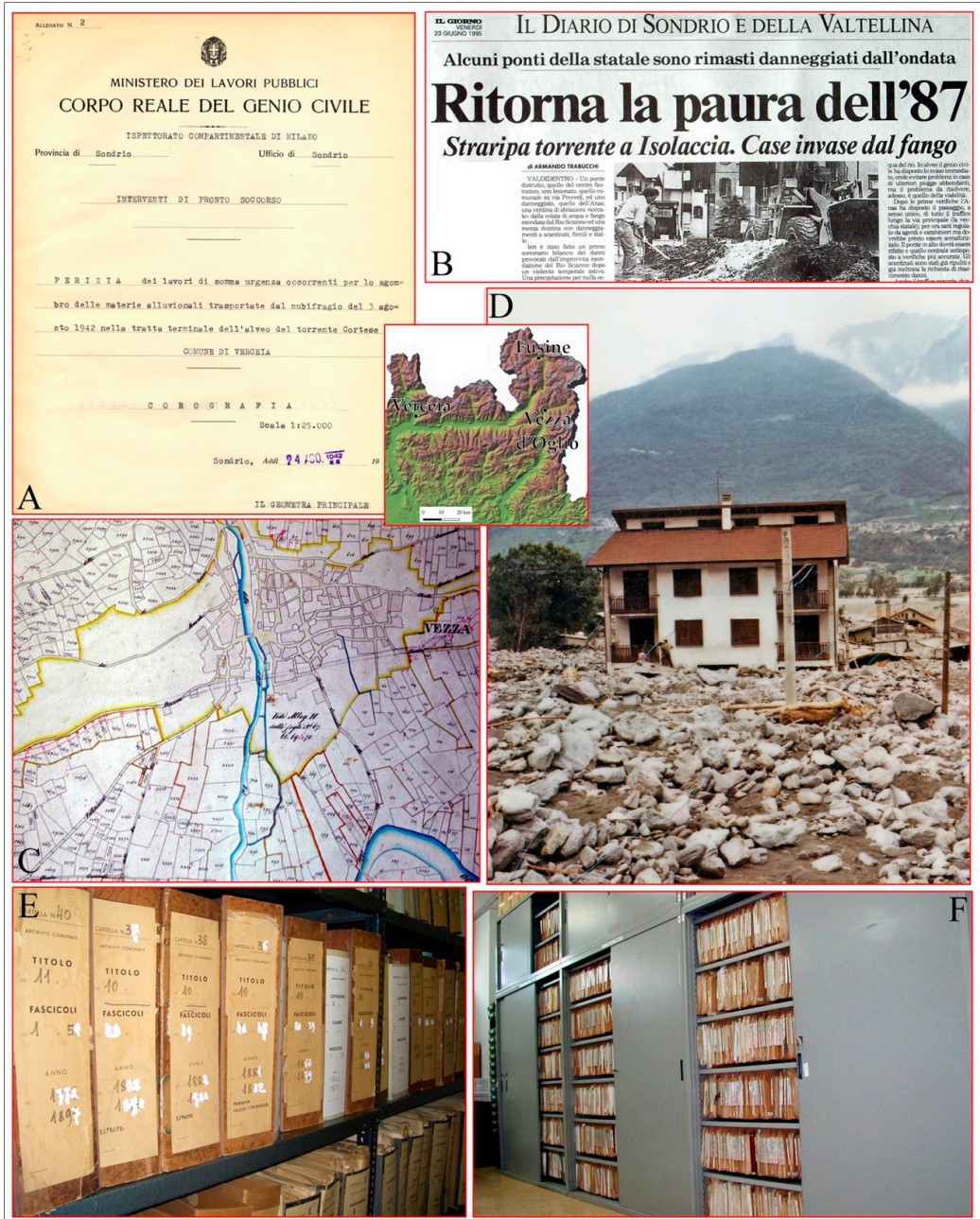

**Figure 4.** Examples of documents collected and used for the historical research of rainfall-induced landslides in the Lombardy Region. (**A**) Municipal archives: report on the damage caused by the debris flows of the Cortese Stream during the event that occurred on 3 August 1942 in the Verceia municipality; (**B**) journalistic sources; news from the "Il Giorno" newspaper on the event of 22 June 1995 in Valtellina; (**C**) State Archives of Brescia: detailed map of the Vezza d'Oglio municipality (Valcamonica) produced during the Kingdom of Lombardy-Venetia (around 1850); (**D**) images collected from local photographers: serious damage in the Fusine town (Valtellina) due to the debris flows of Madrasco Stream on 18 July 1987; (**E**) archive of the Lombardy Region, Sondrio local headquarters: folders containing documents concerning the interventions on the streams made towards the end of the 19th century; (**F**) historical archive of the CNR-IRPI of Turin: the richest Italian archive concerning floods, landslides, and mud-debris flows in Northern Italy.

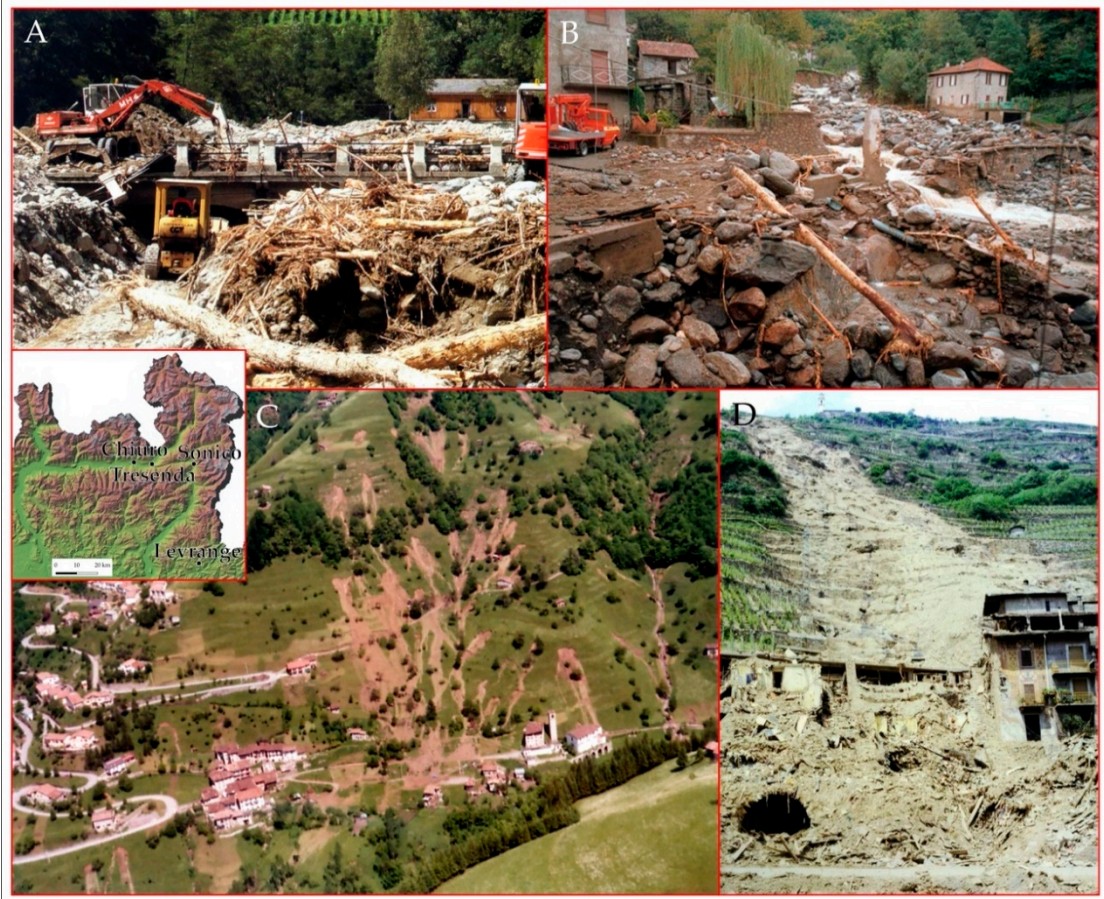

**Figure 5.** Example pf shallow landslides triggered by intense rainfall in the alpine provinces of Lombardy Region: (**A**) view of a bridge totally blocked and overflooded by the exceptional mud-debris flow that occurred at Chiuro, Sondrio Province, on 19 July 1987; (**B**) view of the debris flow that occurred at Sonico, Brescia Province, on 26 September 1987, that destroyed some buildings and one bridge; (**C**) view of widespread soil slips that occurred at Levrange, Brescia Province, on 26 May 1981; (**D**) view of the soil slips triggered on a terraced slope at Tresenda, Sondrio Province, on 22 May 1983, causing severe damage to some buildings.

To obtain an exhaustive catalogue of rainfall-induced landslides, newspapers were most useful to identify the precise or estimated timing of failures, whereas the exact or approximate geographical coordinates were obtained in a Geographical Information System (GIS) using regional base maps at 1:10,000 scale. For each event, the georeferenced database includes: (i) univocal alphanumeric ID; (ii) main watershed and hierarchical higher order catchment within the landslide occurred; (iii) single watercourse affected by the process, particularly in case of mud-debris flows; (iv) location of the area affected by landslides, including administrative and geographical details; (v) data and time (or part of the day, if approximate) of the failure occurrence; (vi) landslide type, according to Hungr et al. [46,47]; (vii) automatic and mechanical rain gauges placed within 5 km from the landslide and type of rainfall measurement available (annual, monthly, daily, hourly, sub-hourly); (viii) spatial accuracy in the geographical location of landslide; (ix) short qualitative description of instability process and ground effects observed on the slopes and/or along the watercourses; (x) consequences of landslides on anthropic structures and activities; (xi) main rock types or soils involved in the failure; and (xii) source of landslide information and/or archive where bibliographic research was carried out.

### 4.2. Research and Validation of Rainfall Data

Once the landslide was geographically located and georeferenced, the spatial analysis was performed to identify the rain gauges closest to the trigger that could be associated with natural instability processes. Once the landslide was geographically located and georeferenced, the spatial analysis was performed to identify the rain gauges that can be associated to the instability processes. Investigation of the relationship between mass movements and precipitations required accurate rainfall measurements of the pluviometric event that have triggered the slope failures. Analysis of rainfall information included: (i) survey of the operative rain gauges networks in the Lombardy Region, (ii) location and selection of the most appropriate gauge correlated to the landslide, and (iii) validation of the accurate hourly measurements available.

For each landslide, rainfall information was obtained from the regional pluviometrical database. The network of rain gauges in Lombardy includes both mechanical (the oldest) and automatic systems that operated for different periods (Figure 6). Mechanical rain gauges were operative at national scale since 1917, then acquired and managed by the Regional Agency for the Environmental Protection (ARPA) since 2005. The ARPA network includes also monitoring sub-networks used for agricultural or meteorological purposes and for hydrological measurements. The regional database contained rainfall data for 56 rain gauges in the Brescia and Sondrio provinces, corresponding to an average density of one rain gauge every 140 km$^2$.

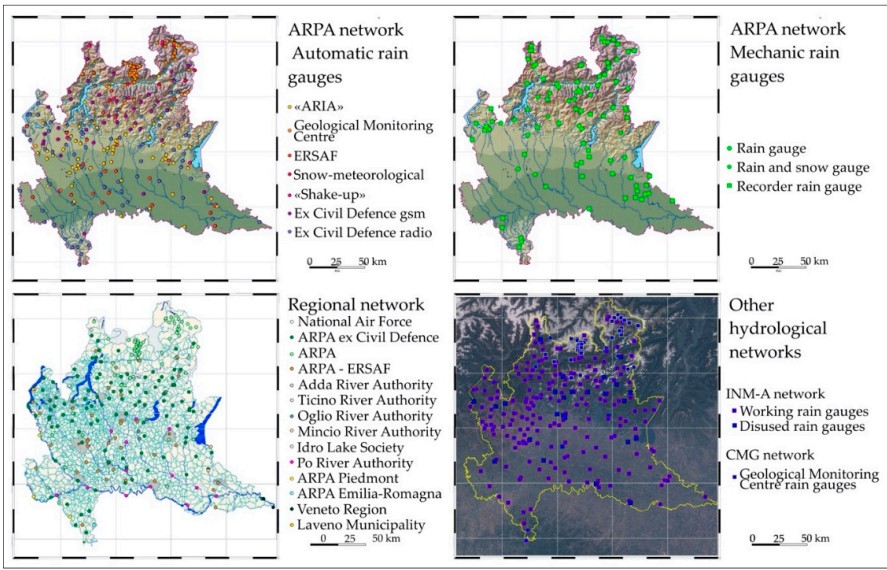

**Figure 6.** Rain gauge networks in the Lombardy Region. ARPA = Regional Agency for the Environmental Protection; ERSAF = Regional Agency for Agricultural and Forestry Services; ARIA = Aria Network consists of instruments installed near cabins belonging to the Air Quality Detection Network; INM-A = Integrated Noise Model Network; CMG = Meteorological automatic acquisition network designed for monitoring some landslides in Lombardy (Sondrio province).

For each individual mass movement for which the time of occurrence was known with sufficient temporal accuracy, we selected the most representative rain gauge. Due to the considerable spatial variations in rainfall regime in the Lombardy Region and the irregular distribution of the hydrological networks, the nearest rain gauge may not be the best option to capture the triggering rainfall: in many cases, they are located in different valleys or catchments or at different elevation (e.g., in the valley floor, on the alluvial fan, etc.), influencing the results of the rainfall threshold analysis. Or, in the case of rain gauges with similar distance from the landslide but located, recorded rainfall measurement may greatly differ from each other, particularly in case of short and very intense localized rainstorms typical of summer months (Figure 7). To avoid errors in rainfall measurements, we selected the most

representative rain gauge considering (i) the location of the rain gauge respect to the geographical and morphological settings, preferably placed in the same catchment as the landslide, (ii) the geographical distance between the rain gauge and the landslides, with a maximum distance of 5 km (±1 km due to spatial errors in gauges and landslides location) based on the spatial distribution of the rain gauges and the geographical and climate settings of the study area, and (iii) the elevation of the rain gauge, compared to the altitude of the landslide.

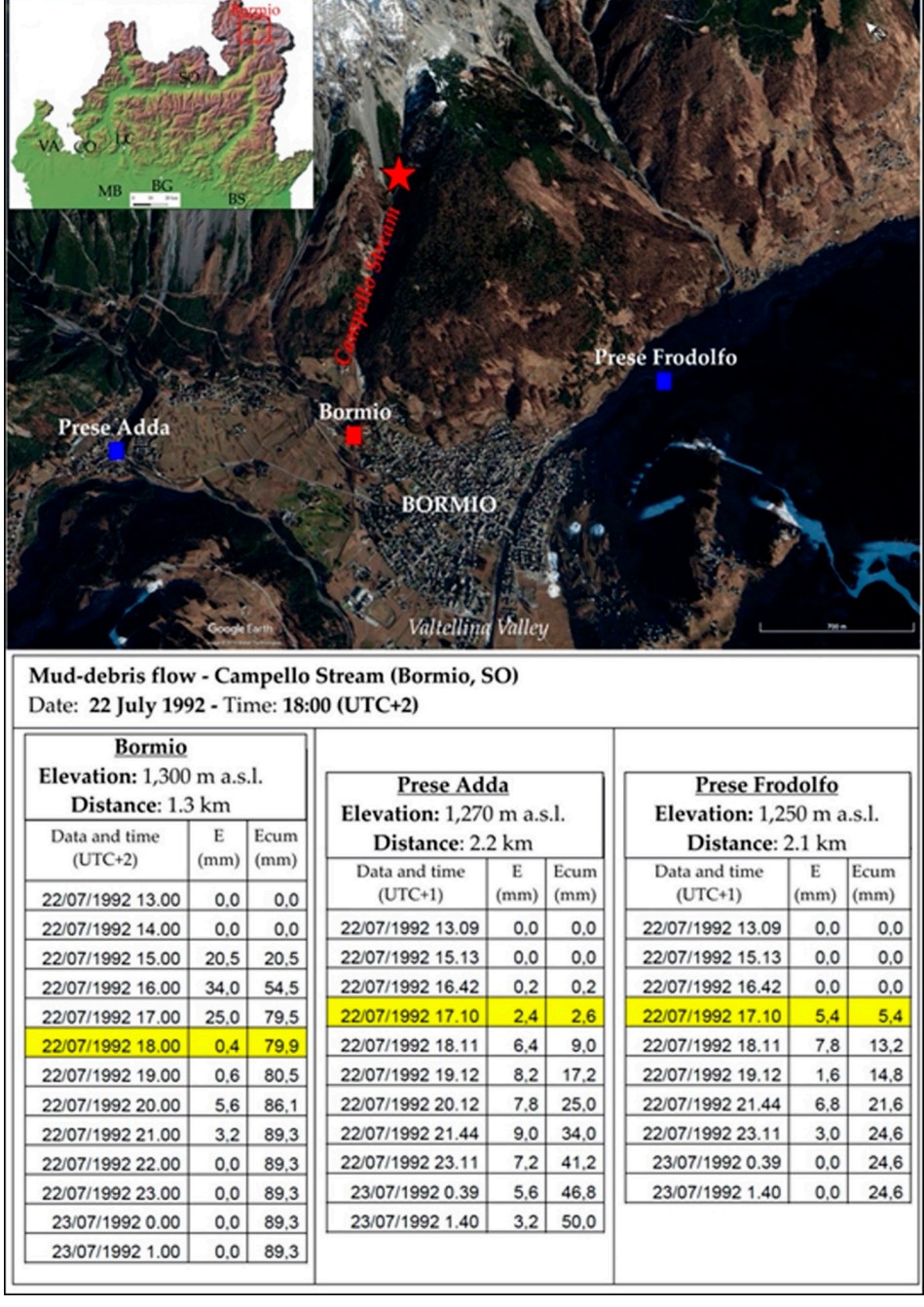

**Mud-debris flow - Campello Stream (Bormio, SO)**
Date: **22 July 1992** - Time: **18:00 (UTC+2)**

| **Bormio** Elevation: 1,300 m a.s.l. Distance: 1.3 km | | | **Prese Adda** Elevation: 1,270 m a.s.l. Distance: 2.2 km | | | **Prese Frodolfo** Elevation: 1,250 m a.s.l. Distance: 2.1 km | | |
|---|---|---|---|---|---|---|---|---|
| Data and time (UTC+2) | E (mm) | Ecum (mm) | Data and time (UTC+1) | E (mm) | Ecum (mm) | Data and time (UTC+1) | E (mm) | Ecum (mm) |
| 22/07/1992 13.00 | 0,0 | 0,0 | | | | | | |
| 22/07/1992 14.00 | 0,0 | 0,0 | 22/07/1992 13.09 | 0,0 | 0,0 | 22/07/1992 13.09 | 0,0 | 0,0 |
| 22/07/1992 15.00 | 20,5 | 20,5 | 22/07/1992 15.13 | 0,0 | 0,0 | 22/07/1992 15.13 | 0,0 | 0,0 |
| 22/07/1992 16.00 | 34,0 | 54,5 | 22/07/1992 16.42 | 0,2 | 0,2 | 22/07/1992 16.42 | 0,0 | 0,0 |
| 22/07/1992 17.00 | 25,0 | 79,5 | 22/07/1992 17.10 | 2,4 | 2,6 | 22/07/1992 17.10 | 5,4 | 5,4 |
| 22/07/1992 18.00 | 0,4 | 79,9 | 22/07/1992 18.11 | 6,4 | 9,0 | 22/07/1992 18.11 | 7,8 | 13,2 |
| 22/07/1992 19.00 | 0,6 | 80,5 | 22/07/1992 19.12 | 8,2 | 17,2 | 22/07/1992 19.12 | 1,6 | 14,8 |
| 22/07/1992 20.00 | 5,6 | 86,1 | 22/07/1992 20.12 | 7,8 | 25,0 | 22/07/1992 21.44 | 6,8 | 21,6 |
| 22/07/1992 21.00 | 3,2 | 89,3 | 22/07/1992 21.44 | 9,0 | 34,0 | 22/07/1992 23.11 | 3,0 | 24,6 |
| 22/07/1992 22.00 | 0,0 | 89,3 | 22/07/1992 23.11 | 7,2 | 41,2 | 23/07/1992 0.39 | 0,0 | 24,6 |
| 22/07/1992 23.00 | 0,0 | 89,3 | 23/07/1992 0.39 | 5,6 | 46,8 | 23/07/1992 1.40 | 0,0 | 24,6 |
| 23/07/1992 0.00 | 0,0 | 89,3 | 23/07/1992 1.40 | 3,2 | 50,0 | | | |
| 23/07/1992 1.00 | 0,0 | 89,3 | | | | | | |

**Figure 7.** Example of variability in rainfall measurements during the short and very intense localized rainstorm that occurred on 22 July 1992 at Bormio (**SO**), which triggered a mud-debris flow along the Campello Stream valley [97]. The three rain gauges are located at similar altitude and distance from the slope failure but a few hundred meters lower than the triggering point and in different sub-catchment. Red star shows the triggering point of mud-debris flow. Squares represent the different type of rain gauge: mechanical (**red**) and automatic (**blue**).

Afterwards, for each rain gauge, we verified the period in which instruments were operative and the availability of hourly rainfall measurements in the different regional networks, in order to determine the rainfall conditions associated with landslides for which geographical and temporal information were known. Even if validate data are usually provided by ARPA and agencies that manage the other meteorological and hydrological networks, we further controlled and validated available datasets with the aim to identify possible gaps in the rain gauge records or lack of homogeneity due to irregular sampling or malfunction in automatic instruments and non-optimal operating conditions in mechanical recorders (e.g., wet or faded graduated paper, pen-nib low ink level, etc.). To further validate pluviometric dataset, we considered a minimum period without rain or "dry period" of 12 h before and after each considered rainfall events.

To define rainfall thresholds, we used measurements recorded by gauges located within 5 km from the landslides for about 50% of the instability processes observed in the 1927–2008 period. Due to the irregular spatial distribution of the regional hydrological network, the elevation of the landslide was not comparable to the altitude of the representative rain gauge. Consequently, errors in rainfall measurements included variations in rainfall regime due to difference in elevation between the landslide and the rain gauges and their geographical distance. This was particularly the case for summer rainstorms that affected restricted mountain areas.

*4.3. Definition of Rainfall Threshold Curves*

For each rainfall condition, we estimated the accumulated event rainfall, E (in mm); the event rainfall duration, $D$ (in h); the average rainfall intensity for the rainfall event, I (in mmh$^{-1}$); and the MAP-normalized mean rainfall intensity, $I_{MAP}$ (in h$^{-1}$). For landslides in which pluviometric measurements from different rain gauges were available, we adopted the station geographically closest to the failures, both in distance and elevation, or, when they were comparable, we adopted the rain gauge that recorded the highest effective rainfall values.

Definition of rainfall duration is essential to identify rainfall thresholds for the occurrence of landslides because it is closely correlated to the mean rainfall intensity and, consequently, to the slope of the curves. In literature, the rainfall duration, $D$ (in h) has been determined in a variety of ways. For example, Brunetti et al. [88] measured the period between the end-time of the rainfall event, coinciding with the time of the landslide, and the start-time of the rainfall event, coinciding with the time when the rain started in the rainfall record. First of all, we measured the period between the time of the landslide and the time when the rain started in the rainfall record preceded by a minimum dry period of 12 h. By analysing the pluviogram recorded by each most representative rain gauge, we observed that short-duration heavy rainfall events, typically in summer months, were generally preceded by dry periods or prolonged very-low intensity precipitations (lower than 1 mm/h). We considered that such low-intensity rainfalls did not affect the initiation of shallow landslides, but they considerably modified the rainfall duration and, consequently, the mean rainfall intensity. Therefore, to include all the rain contributing to the landslide initiation, the rainfall duration has been determined by measuring (i) the period between the time when the rain started in the pluviogram and the time of the landslide occurrence and (ii) a preceding period of minimum 12-h with rainfall intensity lower than 1 mm/h. We considered both conditions to measure rainfall duration, except for some rainfall events extended over a period of several days (more than 5–6 days), typically in spring and autumn months, for which we measured the rainfall duration using only the first condition. In these cases, we observed that the preceding 12-h periods of very-low rainfall intensity (<1 mm/h) were alternated by very heavy showers, which played a not negligible role in triggering landslides.

For landslides for which the rainfall conditions were available, we calculated the rainfall intensity, $I$ (in mm h$^{-1}$) by dividing the accumulated event rainfall in the period considered, *E* (in mm) by the length of the rainfall period, $D$ (in h). Then, we calculated the MAP-normalized rainfall intensity $I_{MAP}$ (in h$^{-1}$) and the normalized intensity rainfall expressed as a percent with respect to MAP $I_N$ (in %) for each most appropriate rain gauge obtained from the rainfall map proposed by Ceriani and Carelli [92].

Using a bi-logarithmic graph, we plotted the rainfall duration and MAP-normalized mean rainfall intensity ($D$, $I_N$) that have resulted in shallow landslides and mud-debris flows in the 1927–2008 period in the Sondrio and Brescia Provinces. We divided summer events, spring-autumn events, and the July 1987 event using different colours.

To define threshold curves, we modelled the distribution of $D$, $I_N$ points associated to rainfall events that resulted in landslides through least-square fitting, and we obtained a linear equation in the bi-logarithmic graph,

$$I_N = \alpha D^{-\beta}, \tag{1}$$

where $I_N$ is the %MAP-normalized mean rainfall intensity (in %), $D$ is the duration of the rainfall event (h), $\alpha$ is the intercept, and $\beta$ defines the slope of the curve. Next, we calculated different curves based on different $\alpha$ values in order to define the corresponding threshold that leaves the 90% of the $D$, $I_N$ points representative of instability phenomena above the fitting curve. Similarly, we defined single curves for the two provinces (Sondrio and Brescia) considered. We established the 90% limit, according to other authors [34], based on several years of experience in the field of floods that hit north-western Italy over the last few decades, in order to limit the maximum number of cases possible and eliminate sporadic and unrepresentative cases, where landslides may have occurred due to other factors, and considering the wide range of rainfall conditions represented in the landslide database. Lastly, we compared the obtained curves to similar empirical thresholds proposed in literature for similar geographical areas, in Italy and worldwide: we adopted a $D/I_N$ diagram in order to evaluate simultaneously three of the most significant parameters that influence the initiation of rainfall-induced landslides (i.e., duration and mean intensity of the rainfall event, MAP in the study area) and compared rainfall conditions in regions with different climatic, geological, and morphological settings. Criteria used by the various authors to define the thresholds have been defined in the section dedicated to the past works and methodologies, when they have been illustrated by authors.

## 5. Results

The historical analysis enabled gathering 1583 rainfall-induced landslides in the Lombardy Region overall, including more than 300 historical events that occurred between the early 16th century to early 20th century. In 900 cases, landslide information were incomplete and/or no quantitative rainfall measurements were available, particularly for events that occurred before 1927. These cases have been included in the catalogue, but we did not use them to define the rainfall threshold curves. Therefore, we obtained accurate landslide and rainfall information for 683 events occurring in the alpine provinces of the Lombardy Region in the 1927–2008 period (Figure 8), for which the exact or approximate location and timing of landslides occurrence were known and the correlated daily or hourly rain measurements were available. In particular, we identified 661 individual mud-debris flows and shallow landslides triggered by rainfall events in the Sondrio and Brescia Provinces.

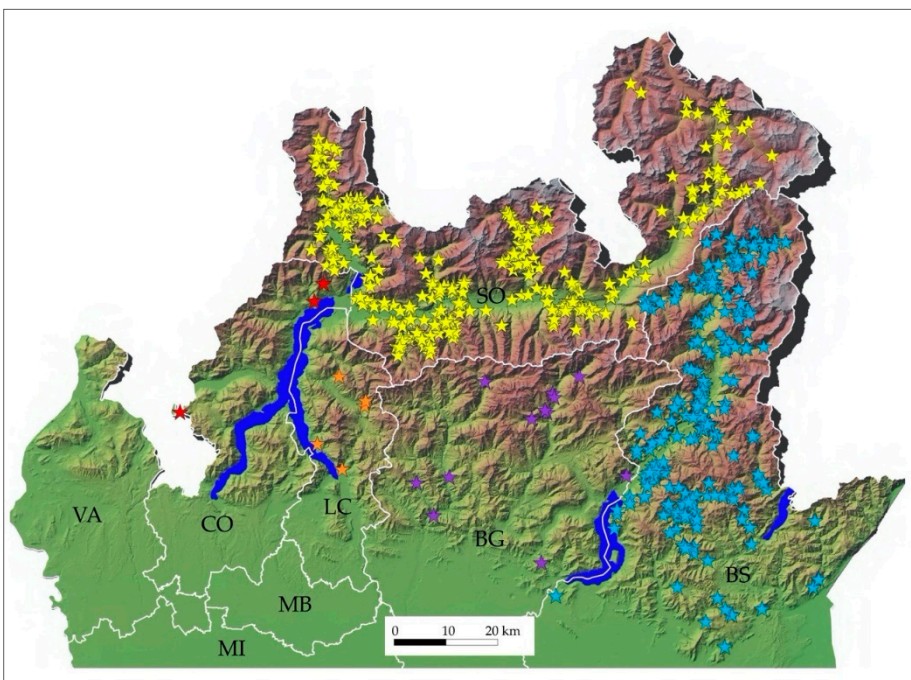

**Figure 8.** Spatial distribution of instability processes (coloured stars) occurred in the alpine and pre-alpine provinces in the Lombardy Region in the period 1927–2008, for which time of failure was known with sufficient temporal accuracy and daily or hourly rainfall measurements. Provinces: BG, Bergamo (**purple**); BS, Brescia (**blue**); CO, Como (**red**); LC, Lecco (**orange**); MB, Monza Brianza; MI, Milano; SO, Sondrio (**yellow**); VA, Varese.

The results of the instability processes analysis in the study area are shown in Figure 9.

Most of the processes (395) were observed in the Sondrio Province, largely between the Valchiavenna and the medium-to-lower Valtellina valleys, whereas landslides in the Brescia Province (266) were well-distributed in the Valcamonica (Figure 9A). Timing of landslide occurrence has been identified with sufficient accuracy in less than 30% of cases: we counted 133 events in the Sondrio Province and 63 events in the Brescia Provinces, for which exact times were available (Figure 9B). On the basis of the type of the rainfall measurements recorded by the most appropriate rain gauge adopted for each event, hourly or sub-hourly data have been available for 52% of the total, with 209 events in the Sondrio Province and 134 in the Brescia Province (Figure 9C). Among the 343 events for which hourly rain data were available, the exact time of occurrence were known for few failures (41% in the Sondrio province and 33% in the Brescia Province, Figure 9C). A further screening of the landslide and rainfall data identified 13 events for which pluviometric measurements were not representative of the rainfall events and/or the timing of occurrence proved to be not reliable. Therefore, they were eliminated.

Finally, we obtained 106 rainfall-induced landslides, including 58 mud-debris flows, 40 shallow landslides, and 8 shallow landslides evolved in flows, for which both hourly rainfall measurements and accurate timing of occurrence were known.

Among the 106 events, we distinguished processes occurred in the spring-autumn (from March to May and from October to November) and in summer months (from June to September): the latter did not include the geo-hydrological events that occurred in July 1987 because their peculiar rainfall conditions were comparable to a typical spring or autumn rainfall event. Analysis of the summer events (Table 1, Figure 10A) revealed that most of the rainfall-induced landslides (54%) occurred at the same time or within 1 h from the maximum recorded peak of rainfall intensity: the percentage increases up to 85% if we consider a period of 3 h from the most intense shower. Only in few cases, failures initiation preceded (8%) or occurred later than 3 h (8%) from the largest rainfall intensity.

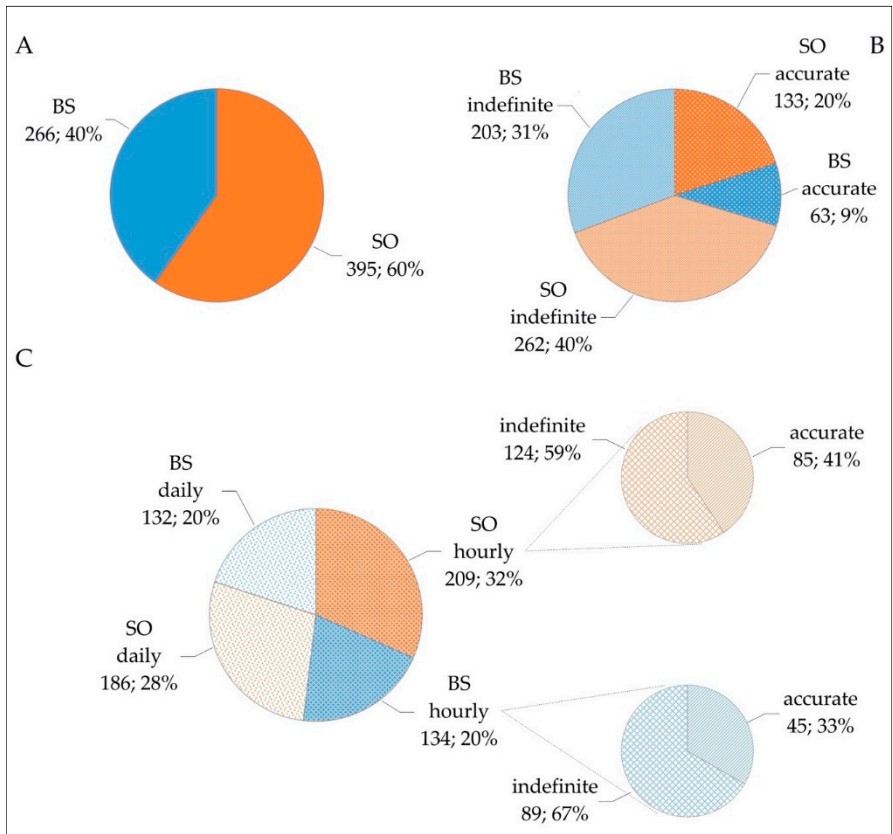

**Figure 9.** Statistical analysis of rainfall induced landslides in the Sondrio and Brescia Provinces in the 1927–2008 period. (**A**) spatial distribution of the processes in the two provinces: SO, Sondrio Province; BS, Brescia Province. (**B**) Classification of the processes based on the accuracy of the time of landslide occurrence (accurate time or indefinite period). (**C**) Classification of the processes based on the type of the rainfall measurements available (hourly or sub-hourly and daily) and further classification of the events for which hourly rainfall data are available based on the accuracy of the time of the landslide occurrence (accurate time or indefinite period).

**Table 1.** Classification of the summer events based on the period (t, in h) between the time of the maximum recorded peak of rainfall intensity and the time of landslide occurrence. For each temporal break, we estimated the number of events (N), the percentage respect the total events (%), and the cumulated percentage ($\%_{cum}$) that represents the percentage of landslides occurred in each temporal break added to the previous ones. $\Delta t < 0$ h represents the period before the maximum rainfall peak.

| $\Delta t$ | 0–1 h | 1–3 h | >3 h | <0 h |
|:---:|:---:|:---:|:---:|:---:|
| N | 21 | 12 | 3 | 3 |
| % | 53.8 | 30.8 | 7.7 | 7.7 |
| $\%_{cum}$ | 53.8 | 84.6 | 92.3 | 100.0 |

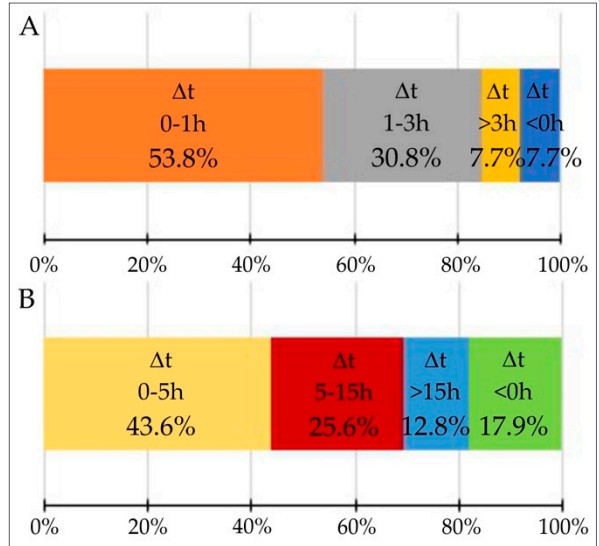

**Figure 10.** Distribution in % of the summer (**A**) and spring-autumn (**B**) events based on the period Δt (in h) between the time of the maximum recorded peak of rainfall intensity and the time of landslide occurrence. Δt < 0 h represents the period before the maximum rainfall peak.

In the spring-autumn events (Table 2, Figure 10B), the period between the trigger of the landslide and the most intense rainfall further differs in duration and distribution: a significant percentage of instability processes triggered at the same time or within 1 to 5 h (44%) or later than 5 h (38%) from the hourly maximum peak of rainfall intensity, whereas in 18% of events, landslide preceded the largest rainfall intensity. We used results of analysis to identify more accurately the timing of triggering landslides when only the date of occurrence was known: adopting a period of 3 h from the maximum peak of rainfall intensity, we further identified 85 summer events for which the part of the day suggested in the information sources match with the most intense phase of the rainfall events. Because of the difficulty of defining the most critical period between the initiation of landslides and the peak of rainfall intensity in the case of the spring-autumn and July 1987 processes, we further identified 86 and 14 events, respectively, for which the available chronological information matched with the most intense showers, based on the correlate hyetographs and the several years of experience in the field of floods that hit north-western Italy over the last decades.

**Table 2.** Classification of the spring-autumn events based on the period (t, in h) between the time of the maximum recorded peak of rainfall intensity and the time of landslide occurrence. For each temporal break, we estimated the number of events (N), the percentage respect the total events (%), and the cumulated percentage (%cum) that represents the percentage of landslides occurred in temporal break interval added to the previous ones. Δt < 0 h represents the antecedent period to maximum rainfall peak.

| Δt | 0–5 h | 5–15 h | >15 h | <0 h |
|---|---|---|---|---|
| N | 17 | 10 | 5 | 7 |
| % | 43.6 | 25.6 | 12.8 | 18.0 |
| %cum | 43.6 | 69.2 | 82.0 | 100.0 |

To define the rainfall conditions that may trigger mud-debris flows and shallow landslides in the study area, we considered a total of 291 rainfall events that occurred in the Sondrio and Brescia Provinces in the period 1927–2008. Mud-debris flows (45%) and shallow landslides (49%) are the most common type of slope failure. For each event, we plotted the event duration, $D$ (in h) and the %MAP-normalized mean rainfall intensity, $I_N$ (in %) values in a bi-logarithmic diagram using different coloured marks for summer, spring-autumn and July 1987 events. We noted that the distribution of

the $(D, I_N)$ empirical points greatly differs with the two classes of rainfall events considered: points representative of spring-autumn events largely converge on duration values higher than 10 h and MAP-normalized intensity values ranging from 0.1% and 0.4%, whereas those representative of summer events were dispersed, including both short and very heavy rainfalls or thunderstorms and prolonged precipitations, over 10 h. Consequently, we singularly processed the rainfall conditions and, finally, obtained two different thresholds that leave 90% of the $(D, I_N)$ empirical points above the curves and intersect for $D = 10$ h.

Coloured lines in Figure 11 represent the thresholds for the possible initiation of landslides in the Central- and South-Eastern Alps sector of the Lombardy Region in the relative ranges of duration:

$$I_N = 0.58D^{-0.46} \ D \leq 10 \text{ h,} \tag{2}$$

$$I_N = 0.31D^{-0.19} \ D > 10 \text{ h.} \tag{3}$$

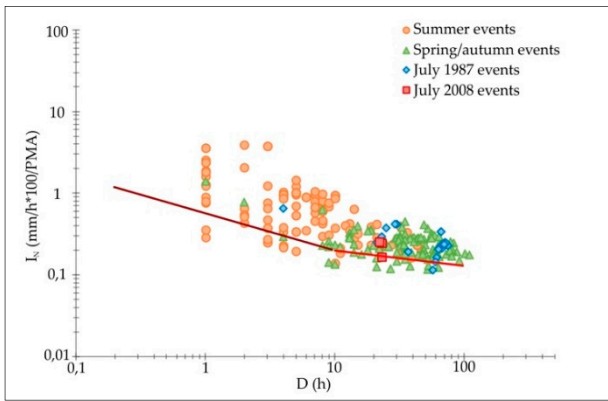

**Figure 11.** Rainfall events (coloured marks) that have resulted in soil slips and mud-debris flows in the Central -and South-Eastern Alps sector of the Lombardy Region in the period 1927–2008. Coloured lines represent threshold that leave 90% empirical points $I_N D$ above the curve for different range of duration (dark red, $D < 10$ h; light red, $D > 10$ h).

Similarly, we singularly processed the $(D, I_N)$ points for each province, and we defined the threshold for the Sondrio Province (Figure 12A):

$$I_N = 0.68D^{-0.50} \ D \leq 10 \text{ h,} \tag{4}$$

$$I_N = 0.32D^{-0.17} \ D > 10 \text{ h,} \tag{5}$$

and the Brescia Province (Figure 12B):

$$I_N = 0.47D^{-0.36} \ D \leq 10 \text{ h,} \tag{6}$$

$$I_N = 0.34D^{-0.22} \ D > 10 \text{ h.} \tag{7}$$

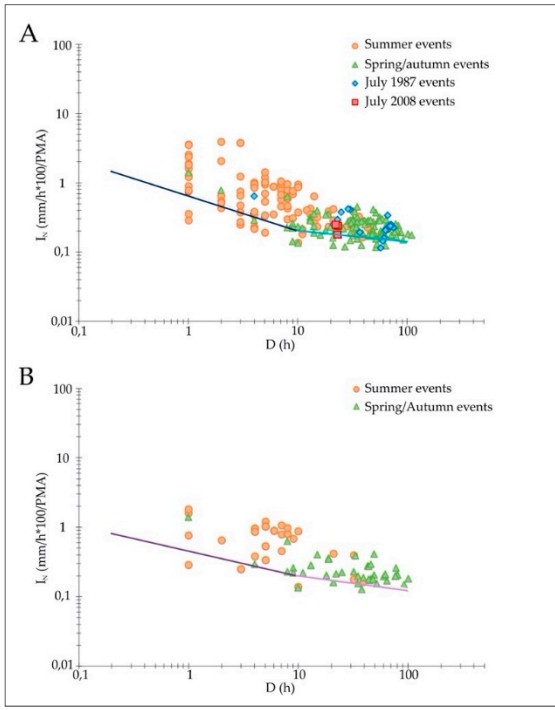

**Figure 12.** Rainfall events (coloured marks) that have resulted in soil slips and mud-debris flows in the Sondrio (**A**) and Brescia (**B**) Provinces in the period 1927–2008. Coloured lines represent the 10% thresholds for different duration ranges (blue and violet, $D < 10$ h; sky-blue and lilac, $D > 10$ h).

## 6. Discussion

Analysis of the rainfall events in the Sondrio and Brescia Provinces between 1927 and 2008 revealed that the rainfall-induced mud-debris flows and shallow landslides occurred primarily: (i) at the same time, or in a period from 1 to 3 h from the maximum recorded peak of rainfall intensity in the summer events and (ii) at the same time or within 5 h (but frequently from 5 to 15 h) in the spring-autumn phenomena. We attribute the different period observed between the most intense rainfall and the timing of landslide occurrence to the different pluviometric features of the rainfall events: short-duration and very intense rainfalls, frequently thunderstorms, in the summer months and prolonged rainfalls, moderate but with abundant cumulative amounts, in the spring and autumn months. Furthermore, we correlate the early occurrence of landslides observed in summer events to prolonged rainfalls characterized by several intense phases that precede the maximum peak rainfall intensity but strong enough to trigger slope failures.

Observation of Figure 13 reveals the range of $D$, $I_N$ conditions likely to result in shallow landslides and mud-debris flows in the Central- and South-Eastern Alps sector of the Lombardy Region and the single Sondrio and Brescia Provinces. We found that the threshold established for the Sondrio Province is higher than the curve obtained for the whole study area and higher to significatively higher than the one defined for the Brescia Province, particularly for small rainfall duration ($D < 10$ h). It means that rainfall intensity required for the initiation of landslides in the Southern and Western Rhaetian Alps sector (Valchiavenna and Valtellina, Sondrio Province) is higher for the same rainfall conditions than those expected in the Lombard Prealps sector (Valcamonica, Brescia Province). For long rainfall duration ($D > 10$ h), the curve established for the alpine sector of the Lombardy Region and the ones for the Brescia Province are comparable.

Curves in Figure 13 are steeper for small rainfall duration ($0.36 < \beta < 0.50$ for $D < 10$ h) than for long duration ($0.17 < \beta < 0.22$ for $D > 10$ h). This highlights a stronger correlation between the duration of the rainfall event and the initiation of the failures for short-duration and very intense rainfall than the prolonged and moderate precipitations typical of spring and autumn months. Results

suggest different meteorological and hydrological conditions for initiation of instability processes within this sector of the Central- and South-Eastern Alps. In the Southern and Western Rhaetian Alps (i.e., Valtellina and Valchiavenna valley and Livigno area), shallow landslides and mud-debris flows can be mostly triggered by high intensity but short duration rainfall events typical of summer months.

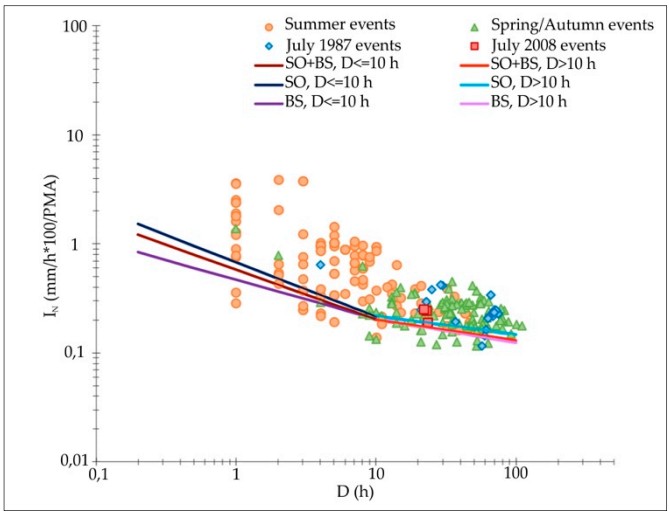

**Figure 13.** Comparison between the rainfall thresholds defined for Central- and South-Eastern Alps sector of the Lombardy Region (SO + BS) and the single Sondrio (SO) and Brescia (BS) Provinces.

Furthermore, we noted that the rainfall conditions that have resulted in mud-debris flows in Valtellina in July 2008 event all plot above the both curves obtained for the Sondrio Province and the whole alpine sector of the Lombardy region.

Adopting a $I_N/D$ diagram, we compared the rainfall thresholds obtained for the Sondrio and Brescia Provinces to empirical curves (Table 3) proposed in the literature for similar alpine provinces, mountain areas, single region, or catchment, in Italy and worldwide (Figure 14).

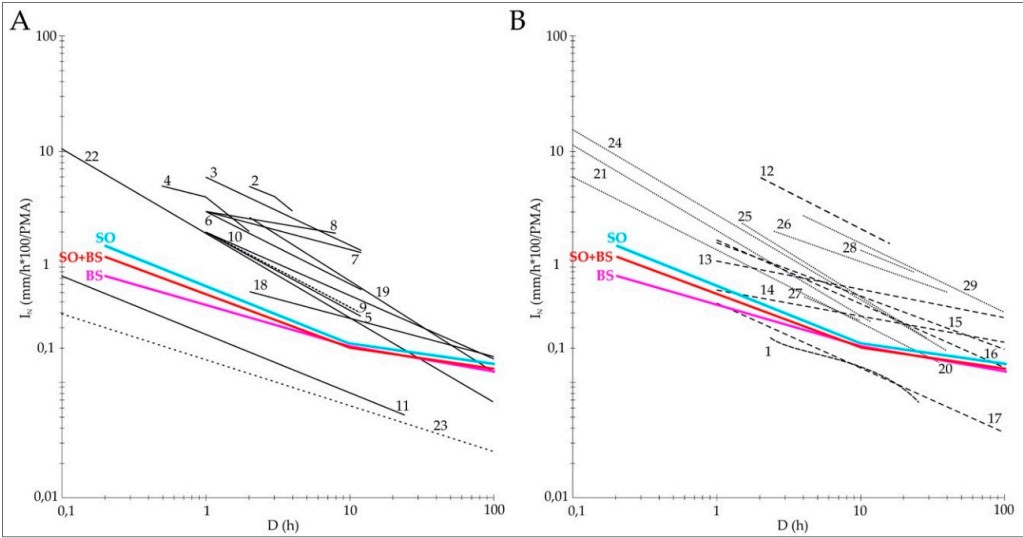

**Figure 14.** Comparison between the rainfall thresholds defined for Central- and South-Eastern Alps sectors of the Lombardy Region (SO + BS, red curve) and the Sondrio (SO, sky-blue curve) and Brescia (BS, fuchsia curve) Provinces and $I_N D$ curves available in literature for similar mountain areas in Italy and worldwide (Table 3): global (broken black curves) and regional (continuous black curves) (**A**) and local (broken black curves) or at single catchment scale (dotted black curves) (**B**).

**Table 3.** MAP-normalized rainfall intensity IMAP ($h^{-1}$)–rainfall duration $D$ (h) thresholds for the initiation of shallow landslides and mud-debris flows in alpine or mountain regions in Italy and worldwide. Geographical extent: G, global; R, regional; L, local; C, catchment. Area: where the threshold was defined. Landslide type: A, all types; D, debris flow; Sh, shallow landslide. Source: 1, Cannon [76]; 2–9, Jibson [77]; 10, Ceriani et al. [67]; 11, Paronuzzi et al. [80]; 12, Wieczorek et al. [82]; 13–16, Aleotti et al. [81]; 17, Bacchini and Zannoni [84]; 18–19, Aleotti [34]; 20–21, Giannecchini [85]; 22, Guzzetti et al. [89]; 23, Guzzetti et al. [69]; 24–26, Giannecchini et al. [87]; 27, Cevasco et al. [90]; 28, Roccati et al. [72]. Criteria used to define the thresholds have been defined in the section dedicated to the past works and methodologies, when they are described or clearly specified by authors.

| # | Extent | Area | Landslide Type | Threshold Curve (I/MAP, $h^{-1}$) | Range (h) |
|---|---|---|---|---|---|
| 1 | L | San Francisco Bay Region, California | D | $D = 46.1 - 3.6 \ 10^3 \ I_{MAP} + 7.4 \ 10^4 \ (I_{MAP})^2$ | $1 < D < 24$ |
| 2 | R | Indonesia | D | $I_{MAP} = 0.07 - 0.01 \ D^1$ | $2 < D < 4$ |
| 3 | R | Puerto Rico | D | $I_{MAP} = 0.06 \ D^{-0.59}$ | $1 < D < 12$ |
| 4 | R | Brazil | D | $I_{MAP} = 0.06 - 0.02 \ D^1$ | $0.5 < D < 2$ |
| 5 | R | Hong Kong | D | $I_{MAP} = 0.02 \ D^{-0.68}$ | $1 < D < 12$ |
| 6 | R | Japan | D | $I_{MAP} = 0.03 \ D^{-0.63}$ | $1 < D < 12$ |
| 7 | R | California | D | $I_{MAP} = 0.03 \ D^{-0.33}$ | $1 < D < 12$ |
| 8 | R | California | D | $I_{MAP} = 0.03 \ D^{-0.21}$ | $0.5 < D < 8$ |
| 9 | G | Worldwide | D | $I_{MAP} = 0.02 \ D^{-0.65}$ | $0.5 < D < 12$ |
| 10 | R | Central Alps, Lombardy, N Italy | D | $I_{MPA} = 0.02 \ D^{-0.55}$ | $1 < D < 100$ |
| 11 | R | Northeastern Alps (Italy) | D | $I_{MAP} = 0.026 \ D^{-0.507}$ | $0.1 < D < 24$ |
| 12 | L | Blue Ridge, Madison County, Virginia | D | $I_{MAP} = 0.09 \ D^{-0.63.}$ | $2 < D < 16$ |
| 13 | L | Sesia Valley, Piedmont, NW Italy | Sh | $I_{MAP} = 1.1122 \ D^{-0.2476}$ | $1 < D < 200$ |
| 14 | L | Ossola Valley, Piedmont, NW Italy | Sh | $I_{MAP} = 0.6222 \ D^{-0.2282}$ | $1 < D < 200$ |
| 15 | L | Lanzo Valleys, Piedmont, NW Italy | Sh | $I_{MAP} = 1.6058 \ D^{-0.4644}$ | $1 > D > 200$ |
| 16 | L | Orco Valley, Piedmont, NW Italy | Sh | $I_{MAP} = 1.6832 \ D^{-0.5533}$ | $1 < D < 200$ |
| 17 | L | Cancia, Dolomites, NE Italy | D | $I_{MAP} = 0.74 \ D^{-0.56}$ | $0.1 < D < 100$ |
| 18 | R | Piedmont, NW Italy | Sh | $I_{MAP} = 0.76 \ D^{-0.33}$ | $2 < D < 150$ |
| 19 | R | Piedmont. NW Italy | Sh | $I_{MAP} = 4.62 \ D^{-0.79}$ | $2 < D < 150$ |
| 20 | C | Apuane Alps, Tuscany (Italy) | Sh | $I_{MAP} = 0.014 \ D^{-0.638}$ | $0.1 < D < 35$ |
| 21 | C | Apuane Alps, Tuscany (Italy) | Sh | $I_{MAP} = 0.0205 \ D^{-0.743}$ | $0.1 < D < 12$ |
| 22 | R | Central and Southern Europe | A | $I_{MAP} = 0.0194 \ D^{-0.73}$ | $0.1 < D < 4000$ |
| 23 | G | Worldwide | Sh, D | $I_{MAP} = 0.0016 \ D^{-0.73}$ | $0.1 < D < 1000$ |
| 24 | C | Middle Serchio Basin, Tuscany (Italy) | Sh | $I_{MAP} = 0.0278 \ D^{-0.74}$ | $2 < D < 40$ |
| 25 | C | Middle Serchio Basin, Tuscany (Italy) | Sh | $I_{MAP} = 0.0325 \ D^{-0.78}$ | $1.5 < D < 40$ |
| 26 | C | Middle Serchio Basin, Tuscany (Italy) | Sh | $I_{MAP} = 0.0301 \ D^{-0.44}$ | $2.5 < D < 40$ |
| 27 | C | Bisagno catchment, Liguria (Italy) | A | $I_{MAP} = 0.0112 \ D^{-0.525}$ $I_{MAP} = 0.0461 \ D^{-0.525}$ | $4 < D < 10 \ 10 < D < 24$ |
| 28 | C | Entella Basin, Liguria (Italy) | Sh, D | $I_{MAP} = 0.063 \ D^{-0.60}$ | $4 < D < 169$ |

The new rainfall thresholds are generally lower to significantly lower than curves defined for other mountain areas meaning that a lower rainfall mean intensity is required to initiate shallow landslides and mud-debris flows in the Central- and South-Eastern Alps sector of the Lombardy region,

than in other alpine sectors and mountain regions. Furthermore, it means that we defined critical rainfall conditions more precautionary than those defined by other authors. Specifically, for short rainfall duration ($D$ < 10 h), the new thresholds are substantially lower than the global and regional curves proposed by Jibson [77] for different mountain areas around the world and the regional curves proposed by (i) Ceriani et al. [67] for the Lombardy Region, (ii) Aleotti [34] for the Piedmont region, and (iii) Guzzetti et al. [89] for the South and Central Europe. They are higher, instead, than the curve proposed for the northeastern Italian Alps by Paronuzzi et al. [80] and the global curve defined by Guzzetti et al. [69] (Figure 13). The thresholds obtained for the alpine provinces in the Lombardy region are also substantially lower than the local curves proposed by (i) Wieczoreck et al. [82] for the Blue Ridge are in Virginia, (ii) Aleotti et al. [81] for some neighbouring valleys in the Eastern and Central Alps (i.e., Sesia, Lanzo and Orco valleys), and (iii) Giannecchini [85] for the Apuane Alps, in Tuscany. The new thresholds are also lower than some curves defined for single mountain catchment in Italy, such as the curves proposed by (i) Giannecchini et al. [87] for the Serchio river, in Tuscany, (ii) Cevasco et al. [90] for the Bisagno stream, and (iii) Roccati et al. [72] for the Entella river in the Liguria region. The new thresholds are, instead, similar to the curves proposed by Bacchini et Zannoni [84] for the Cancia catchment, in the Dolomites, and by Aleotti et al. [81] for the Ossola valley in Piedmont (Figure 13).

For long rainfall duration ($D$ > 10 h), comparison between the $I_N/D$ thresholds reveals a gradual convergence between the curves, in particular for duration rainfall exceeding 50 h (Figure 14): for example, the regional curves proposed by Aleotti [34] and Guzzetti et al. [89] (Figure 14A) and some local thresholds defined by Giannecchini [85], Giannecchini et al. [87], and Aleotti et al. [81] (Figure 14B). Furthermore, we noticed that the new thresholds obtained for the Sondrio and Brescia Provinces are generally gentler ($0.17 < \beta < 0.22$, for $D$ > 10 h) than those obtained for other mountain areas in Italy and worldwide ($0.21 < \beta < 0.79$).

Even if normalization to the MAP allows us to compare areas with different meteorological settings, we noticed a large variability between the curves considered. According to the results obtained by Govi et al. [73], Giannecchini [86], and Giannecchini et al. [87], the intensity rainfall values needed to initiate shallow landslides in the Sondrio and Brescia Provinces are lower than those observed in other mountain areas because of the lower MAP values that characterize the study area: for example, the curves established for the Apuane Alps (MAP = 1870 mm/y) by Giannecchini [85], the Serchio catchment (MAP = 1565 mm/y), or the Entella catchment (MAP = 1800 mm/y) by Roccati et al. [72] are substantially higher than the curves defined for the Sondrio and Brescia Provinces, characterized by MAP values ranging from 650 mm/y to 1300 mm/y in large sectors of Valtellina and Valcamonica.

We attribute the results mainly to the type of critical geo-hydrological events and, secondarily, to geological, geomorphological, land-use, and meteorological settings of the Central- and South-Eastern Alps. However, we note that the different criteria adopted to identify the rainfall events and define the critical rainfall duration values, in addition to the different approaches used to establish the rainfall thresholds and the different type of rainfall and landslides information analysed introduce possible uncertainty in the comparison with other curves.

The analysis of the rainfall-induced landslides that occurred in the Lombardy Region furthermore revealed some important issues to take into account in the study concerning the rainfall thresholds: (i) selection of validate rain measurements, (ii) definition of the critical rainfall duration responsible for the initiation of landslides, (iii) selection of the most appropriate rain gauge based on the distance from the landslide and the type of rainfall events (e.g., prolonged and widespread processes typical of spring-autumn months or short-duration and very localized rainstorms very frequently in summer), and (iv) spatial variability of mean annual precipitation (MAP) and rainfall intensity and cumulate of a single event, also correlated to altitude.

## 7. Conclusions

Historical analysis of the geo-hydrological processes in the Lombardy region has identified 683 rainfall-induced landslides in the 1927–2008 period for which mass movements and rainfall information were known with a enough geographical and temporal accuracy. Almost all of the shallow landslides and mud-debris flows (661) were observed in the Valtellina and Valcamonica valleys, in the Central- and South-Eastern Alps, compared to the Sondrio (395) and Brescia (226) Provinces. Among these 661 events, we used rainfall and mass movements information for 291 rainfall events, for which validated hourly rain measurements were available, to define new rainfall MAP-normalized intensity–duration ($I_ND$) thresholds for the possible initiation of shallow landslides and mud-debris flows in the alpine districts of the Lombardy Region. We established three different thresholds that leave 90% of the $I_ND$ points above the curves: a large-scale threshold for the whole Central- and South-Eastern Alps sector and two local curves for the Sondrio (Valtellina, Southern and Western Rhaetian Alps and Central Lombard Prealps) and the Brescia (Valcamonica, Central and Eastern Lombard Prealps) Provinces, respectively.

We found that the threshold established for the Sondrio Province is higher and slightly steeper than the curve obtained for both the Brescia Province and the regional alpine sector, particularly for small rainfall duration, $D < 10$ h. We attribute the result mainly to the geological, geomorphological, orographic, and meteorological settings of the Valtellina valley. Comparison of the new rainfall thresholds to other empirical curves proposed in literature for similar alpine regions, local mountain areas, or single regions or catchments in Italy and worldwide revealed that our thresholds are generally lower, mainly due to the type of rainfall events used for the analysis and, secondarily, to the geological, geomorphological, orographic, and meteorological settings of the Sondrio and Brescia Provinces that greatly increase the susceptibility to rainfall-induced shallow landsliding.

Analysis of the rainfall events also allowed us to recognize a period ranging from less than 1 to 3 h (summer events) or 5 h (spring-autumn events) for the expected initiation of shallow landslides and mud-debris flows. Furthermore, historical analysis highlights that most of the areas characterized today as having a high landslide risk have been historically affected by destructive mud-debris flows or shallow landslides and are where anthropogenic modification has frequently increased the risk due to land-use changes and urbanization.

Our study provides to the international literature a further contribution to the knowledge framework on shallow landslides and mud-debris flows and their expected timing of occurrence in the alpine districts, where the presence of structures and infrastructures on slopes and alluvial fans further increase the landslide risk. It is important to underline how these natural processes caused severe damage to population and property in the past and continue to do so every year. In this terms, rainfall thresholds established for the Central- and South-Eastern Alps in the Lombardy region represent a useful tool for the Civil Protection to improve the existent early-warning system in order to reduce the "false negative" or "missed alerts", which unnecessarily alarm the population with a consequent increasing mistrust of the authorities in charge of risk management and governance, and support the best strategies to mitigate landslide risk. The understanding of potentially dangerous and destructive mass movements, such as mud-debris flows and shallow landslides, and the knowledge of historical processes are essential to identify high susceptibility areas and assess the possible scenarios also related to the effects of climate changes, in order to support decision-makers and stakeholders in sustainable land-use and planning and landslides risk management.

The scientific approach used was very rigorous and led to the exclusion of much data that did not meet the required requirements, in order to reach an objective, reliable, verifiable, and shareable reality knowledge.

The approach proposed can be adopted in similar high risk scenarios in alpine and mountain environments to define critical rainfall conditions that result in mud-debris flows and shallow landslides, in order to develop a comprehensive early-warning system and adopt the best practices to mitigate the landslide risk and reduce damage such natural processes may cause to population and property.

**Author Contributions:** All authors contributed to the research presented in this work. Their contributions are presented below. Conceptualization, F.L.; Methodology, F.L.; Software, A.R.; Validation, A.R., M.B. and C.G.C.; Investigation, M.B., C.G.C. and A.R.; Resources, F.L.; Data curation, M.B., C.G.C. and A.R.; Writing—original draft preparation, F.L., J.D.G., M.B., C.G.C., F.F., A.R. and L.T.; Writing—review and editing, F.L., J.D., M.B., C.G.C., F.F., A.R. and L.T.; Visualization, F.F. and L.T.; Supervision, F.L. and J.D.G.; Project administration, L.T.; Funding acquisition, F.L., F.F. and L.T. Each author has approved the submitted version and agrees to be personally accountable for the author's own contributions and for ensuring that questions related to the accuracy or integrity of any part of the work, even ones in which the author was not personally involved, are appropriately investigated, resolved, and documented in the literature. All authors have read and agreed to the published version of the manuscript.

**Funding:** This research was funded by ARTEMIDE Project (Archive ThEMatic Imagin Aerophotograf of Events) (http://www.irpi.cnr.it/en/project/artemide/), supported by Compagnia di San Paolo Italian National Foundation.

**Conflicts of Interest:** The authors declare no conflict of interest.

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
