# Peer review of "Eighty Years of Data Collected for the Determination of Rainfall Threshold Triggering Shallow Landslides and Mud-Debris Flows in the Alps"

_water, doi:10.3390/w12010133_

Round 1

Reviewer 1 Report

This study tried to establish a rainfall threshold for shallow landslides and mud-debris flows of the northern mountains of Italy. The authors collected 661 rainfall-induced landslides from 1927 to 2008 in the Valtellina and Valcamonica valleys to establish a new threshold rainfall intensity. Among them, only 291 events had hourly rain measurements and were used to define the rainfall intensity (normalized by the mean annual precipitation, MAP) vs. duration threshold lines for the Sondrio Province (SO), Brescia Province (BS), and the two regions combined (SO + BS). The authors also compared the new threshold lines with those found in the literature for similar mountainous areas. They discovered that the new threshold lines were lower than most of the existing threshold lines, perhaps because they were more precautionary (line 582) than the other authors in the literature. The manuscript is well written and has a very comprehensive data collection and analysis. However, some of the data presentations are not as ideal as they could be:

First, this study only considered the rainfall duration and the MAP-normalized mean rainfall intensity as the parameters of the rainfall threshold. No other parameters were considered (such as antecedent precipitation amounts and absolute rainfall intensities). However, this seems to be a common approach in this part of the world. Nevertheless, I think the authors need to clearly define the rainfall duration (D) in this study to avoid any misinterpretation of the results. For example, the authors said in lines 407-409 that "Whereas, in cases of prolonged rainfall events with several showers, typical in spring and autumn months, we considered a further 12-hours antecedent period with high intensity rainfall in order to include all the rain contributing to the landslide initiation." Did they mean that they added 12 hours to the rainfall duration (D)? Similarly, did they add a 12-hour dry period (line 405) to the rainfall duration (D) or simply use the dry period to define the start of a rainfall?

Second, the authors did not explain why this study (threshold lines) was based on a 90% chance of exceedance. Why not any other numbers (such as 95%, 85%, and so on)? The authors admitted that they were more precautionary than the other authors in the literature (line 582). Then, what numbers did the other authors use in the literature? Furthermore, if the numbers are different, how could the authors of the manuscript compare their results to the literature? Specifically, are the references quoted in Table 3 all based on a 90% chance of exceedance as in this study? I suggest the authors should add the definition of thresholds of different studies to Table 3.

Third, the authors compared the threshold lines obtained in this study with other threshold lines in the literature. However, there was only one global reference (which was published in 1989), and the rest were only from catchments and local/regional scales (see Table 3). Ideally, more references would be preferred.

Finally, some part of the manuscript seems to be put together hastily. For example, the "%cum" numbers in Table 2 were all wrong. They should have been 43.6, 69.2, 82.0, and 99.9. Also, Figure 8 needs quite some revising:

- The total numbers (number of rainfall-induced landslides) in the pie charts are different in Figure 8A (291), Figure 8B (621), Figure 8C (661), and Figure 8D (661). This is very confusing given that these pie charts are grouped together in the same figure. The authors did not explain the discrepancy in the total numbers in the caption of Figure 8 or the related text. I did not understand where the number 291 came from until I read line 514, which was already two pages past Figure 8.

- The sub-figure of Figure 8D (lower right-hand corner) was labeled wrong. Both legends read "accurate." Also, they added up to 102 (34 + 68) instead of 134. Why? There seems to be something wrong.

Here are a few additional comments:

- Is IN the same as IMAP? Please define them clearly.

- The use of the label "< 0h" in Figure 9, Table 1, and Table 2 is a little bit odd. I suggest the authors should add some explanations to the figure and the tables. (What does < 0h mean?)

- The caption of Figure 11 said 5% threshold. I think it should have been 10% according to the context of the manuscript.

- The statement in lines 563-565 was wrong. There were points of the July 2008 event plotted on and below the threshold lines in Figure 11.

- Figure 6: Fonts are too small to be legible. A sans-serif typeface such as Helvetica and Arial is preferred.

- Lines 25-27 (abstract): "Furthermore, we found that landslides occurred primarily … in a period from 5 to 15 hours in spring-autumn events." This is a wrong statement. According to Table 2, it should have been 0-5 hours.

- Line 162: Did you mean "MAP"-normalized rainfall thresholds?

- Line 377: … temporal information "were" known.

- Line 477: Mai is a typo.

- Line 480: It should have been Figure 9B.

- Line 493: It should have been Figure 9A.

Author Response

Review Report Form 

Open Review

English language and style

( ) Extensive editing of English language and style required  
( ) Moderate English changes required  
(x) English language and style are fine/minor spell check required  
( ) I don't feel qualified to judge about the English language and style  

Yes

Can be improved

Must be improved

Not applicable

Does the introduction provide sufficient background and include all relevant references?

(x)

( )

( )

( )

Is the research design appropriate?

(x)

( )

( )

( )

Are the methods adequately described?

( )

(x)

( )

( )

Are the results clearly presented?

( )

(x)

( )

( )

Are the conclusions supported by the results?

( )

(x)

( )

( )

Comments and Suggestions for Authors

This study tried to establish a rainfall threshold for shallow landslides and mud-debris flows of the northern mountains of Italy. The authors collected 661 rainfall-induced landslides from 1927 to 2008 in the Valtellina and Valcamonica valleys to establish a new threshold rainfall intensity. Among them, only 291 events had hourly rain measurements and were used to define the rainfall intensity (normalized by the mean annual precipitation, MAP) vs. duration threshold lines for the Sondrio Province (SO), Brescia Province (BS), and the two regions combined (SO + BS). The authors also compared the new threshold lines with those found in the literature for similar mountainous areas. They discovered that the new threshold lines were lower than most of the existing threshold lines, perhaps because they were more precautionary (line 582) than the other authors in the literature. The manuscript is well written and has a very comprehensive data collection and analysis. However, some of the data presentations are not as ideal as they could be:

First, this study only considered the rainfall duration and the MAP-normalized mean rainfall intensity as the parameters of the rainfall threshold. No other parameters were considered (such as antecedent precipitation amounts and absolute rainfall intensities). However, this seems to be a common approach in this part of the world. Nevertheless, I think the authors need to clearly define the rainfall duration (D) in this study to avoid any misinterpretation of the results. For example, the authors said in lines 407-409 that "Whereas, in cases of prolonged rainfall events with several showers, typical in spring and autumn months, we considered a further 12-hours antecedent period with high intensity rainfall in order to include all the rain contributing to the landslide initiation." Did they mean that they added 12 hours to the rainfall duration (D)? Similarly, did they add a 12-hour dry period (line 405) to the rainfall duration (D) or simply use the dry period to define the start of a rainfall?

We clarified how we determine the rainfall duration in the study to avoid misinterpretation of the results, as suggested. By analysing the pluviogram recorded by each most representative rain gauge, we to determine the rainfall duration by measuring i) the period between the time when the rain started in the pluviogram and the time of the landslide occurrence and ii) a preceding period of minimum 12-hours with rainfall intensity lower than 1 mm/h. We considered both conditions to measure rainfall duration, except for some rainfall events extended over a several days period (more than 5-6 days), typically in spring and autumn months, for which we measured the rainfall duration using only the first condition. In these cases, we observed that the preceding 12-hours periods of very-low rainfall intensity (< 1 mm/h) were alternated by very heavy showers, which played a not negligible role in triggering landslides.

Second, the authors did not explain why this study (threshold lines) was based on a 90% chance of exceedance. Why not any other numbers (such as 95%, 85%, and so on)? The authors admitted that they were more precautionary than the other authors in the literature (line 582). Then, what numbers did the other authors use in the literature? Furthermore, if the numbers are different, how could the authors of the manuscript compare their results to the literature? Specifically, are the references quoted in Table 3 all based on a 90% chance of exceedance as in this study? I suggest the authors should add the definition of thresholds of different studies to Table 3.

In some disciplines (health, for instance), it would not be accepted an alpha under 0.01 (99%). Although in the social sciences, despite typically adopting 95% (0.05), it is possible to accept a 90% (0.10) as well. In the present study, where the concurrence of many of many factors, besides rainfall, can result in occurrence of landslide, and due to the wide range of rainfall conditions included in the database, 90% was considered acceptable by authors, and set by the research group after a long speech with one of the maximum Italina experts in the field of soil slips (Prof. Mario Govi). In fact, many previous studies delineated thresholds that enveloped the totality of landslide-inducing rainfall, but many considered limited number of events in al limited period.

Besides, other authors established arbitrarily the 90% limit based on the need to both limit the maximum number of cases possible and to eliminate sporadic and unrepresentative cases (Aleotti, 2004).

As related by Guzzetti et al. (2008), “a threshold is the minimum or maximum level of some quantity needed for a process to take place or a state to change (White et al., 1996). A minimum threshold defines the lowest level below which a process does not occur. A maximum threshold represents the level above which a process always occurs. For rainfall-induced landslides a threshold may define the rainfall, soil moisture, or hydrological conditions that, when reached or exceeded, are likely to trigger landslides. Rainfall thresholds can be defined on physical (process-based, conceptual) or empirical (historical, statistical) bases (Corominas, 2000; Crosta and Frattini, 2001; Aleotti, 2004; Wieczorek and Glade, 2005, and references therein)”. According to Segoni et al. (2017) “a threshold represents the lower bound of known hydrological conditions (e.g., rainfall, infiltration, soil moisture) that resulted in landslides (Reichenbach et al. 1998). In a Cartesian plane, thresholds are expressed in terms of curves that delimit a portion of the plane containing the hydrological conditions related to known slope failures. […]  A further improvement consists in dividing the Cartesian plane in three parts, by means of two thresholds: a lower threshold, below which no landslides occur, and an upper threshold, above which landslides always occur (Wilson et al. 1993). Between the two thresholds, different probabilities of occurrence are defined, with uncertainties related to the incompleteness of knowledge on the physical process (Crozier 1997) and on the landslide database.”. Therefore, in literature different approaches and criteria to establish rainfall thresholds have been adopted, based on the features of the landslide database used in the analysis and the aims of the works. Since the objective of a threshold is to separate rainfall conditions leading to (at least) instability (above the threshold) and stability (below the threshold), in literature both single and multiple curves are depicted, particularly in case of complex thresholds systems used in landslide warning systems  where different levels of warning can correspond to the exceeding of different thresholds. In our paper, we considered MAP-normalized mean intensity rainfall-rainfall duration curves defined for similar geographical areas, which are defined adopted different empirical approaches (statistical, probabilistic, manual) and the criteria to drawing the curve (e.g., the number of the exceedance level) are frequently not described or not clearly specified. It was not possible to add definition of thresholds of different studies to Table 3, because for many studies this information is missing in their original sources or not clearly specified. When possible, for thresholds listed in table 3 (Ceriani et al., 1994; Aleotti et al., 2002; Aleotti, 2004; Bacchini and Zannoni, 2003; Giannecchini, 2005; Giannecchini et al., 2012; Roccati et al., 2018), thresholds have been already briefly defined in section 2; in order to respond to the reviewer’s query, we further better defined the thresholds in section 2 and implemented the definition of the other curves listed in table 3

Bibliography:

Hair Jr., J.F., Black, W.C., Babin, B.J. and Anderson, R.E. (2009). Multivariate Data Analysis. 7th Edition, Prentice Hall, Upper Saddle River, NJ. Hazelrigg, L. (2009). Inference. In: M. Hardy and A. Bryman (eds.). The Handbook of Data Analysis. Sage, London, UK. White ID, Mottershead DN, Harrison JJ (1996) Environmental Systems, 2nd Edition. London: Chapman & Hall, 616 pp Corominas J (2000) Landslides and climate. Keynote lecture- In: Proceedings 8th International Symposium on Landslides, (Bromhead E, Dixon N, Ibsen ML, eds). Cardiff: A.A. Balkema, 4: 1–33 Crosta GB, Frattini P (2001) Rainfall thresholds for triggering soil slips and debris flow. In: Proceedings 2nd EGS Plinius Conference on Mediterranean Storms (Mugnai A, Guzzetti F, Roth G, eds). Siena: 463– 487 Aleotti P (2004) A warning system for rainfall-induced shallow failures. Eng Geol 73: 247–265 Wieczorek GF, Glade T (2005) Climatic factors influencing occurrence of debris flows. In: Debris flow Hazards and Related Phenomena (Jakob M, Hungr O, eds). Springer Berlin Heidelberg, 325–362 Reichenbach P, Cardinali M, De Vita P, Guzzetti F (1998) Hydrological thresholds for landslides and floods in the Tiber River basin (central Italy). Environ Geol 35:146–159. Crozier MJ (1997) The climate-landslide couple: a southern hemisphere perspective. In: Matthews JA, Brunsden D, Frenzel B, Gläser B, Weiß MM (eds) Rapid mass movement as a source of climatic evidence for the Holocene. Gustav Fischer, Stuttgart, pp 333–354

Third, the authors compared the threshold lines obtained in this study with other threshold lines in the literature. However, there was only one global reference (which was published in 1989), and the rest were only from catchments and local/regional scales (see Table 3). Ideally, more references would be preferred.

To compare the rainfall thresholds established for the study area with other curves in literature, we adopted only similar IMAP-D thresholds, with a particular attention to similar alpine and mountain regions.

Most of rainfall thresholds based on MAP-normalized intensity and rainfall duration are defined for local/regional areas or single catchment (see also the world-wide database of empirical rainfall thresholds for the possible occurrence of landslides compiled by CNR-IRPI available at: http://wwwdb.gndci.cnr.it/php2/rainfall_thresholds/thresholds_type.php?typesigla=IMAP-D&lingua=it)

In Table 3 and correlate Figure 13, we implemented a further global reference defined by Guzzetti et al., (2008) for shallow landslides and debris flows.

Finally, some part of the manuscript seems to be put together hastily. For example, the "%cum" numbers in Table 2 were all wrong. They should have been 43.6, 69.2, 82.0, and 99.9

 “%cum” numbers in Table 2 have been corrected.

Also, Figure 8 needs quite some revising the total numbers (number of rainfall-induced landslides) in the pie charts are different in Figure 8A (291), Figure 8B (621), Figure 8C (661), and Figure 8D (661). This is very confusing given that these pie charts are grouped together in the same figure. The authors did not explain the discrepancy in the total numbers in the caption of Figure 8 or the related text. I did not understand where the number 291 came from until I read line 514, which was already two pages past Figure 8.

Numbers of rainfall-induced landslides in the pie-charts in the sub-figure 8B, 8C and 8D have been revised and now the total numbers match each other (now Figure 9).

The pie-chart in the sub-figure A was correlated to the subset of events (291) used to define the thresholds: since we explained how we obtained this number a few paragraphs past, we preferred to removed the statistical data of landslide type (MDF, SS, MDF/SS) from here and we put them in the paragraph where we explained where the number 291 came from. For the same reason, we preferred to remove the pi-chart in sub-figure A to avoid to give rise to misunderstandings.

The sub-figure of Figure 8D (lower right-hand corner) was labeled wrong. Both legends read "accurate." Also, they added up to 102 (34 + 68) instead of 134. Why? There seems to be something wrong.

Labels in sub-figure D have been adjusted. Numbers in last pie-chart (those correlated to BS) contained one error, we apologized for the mistake. We corrected the numbers and now they correctly added up to 134.

 Here are a few additional comments:

Is INthe same as IMAP? Please define them clearly.

IMAP is the MAP-normalized rainfall intensity (in h-1); IN is the normalized rainfall intensity expressed as a percent with respect to MAP (in %), as defined by Bacchini and Zannoni (2003).

We defined more clearly IMAP and IN to avoid any confusion.

The use of the label "< 0h" in Figure 9, Table 1, and Table 2 is a little bit odd. I suggest the authors should add some explanations to the figure and the tables. (What does < 0h mean?)

The label “< 0h”, i.e., a negative lag time between the landslide initiation and the maximum recorded peak of rainfall intensity, means that the landslide occurred before the maximum peak of rainfall intensity.  We adjusted the labels in Figure 9 (now Figure 10) and Tables 1,2 and we added some explanations to its meaning in the text and in the captions to clarify it.

The caption of Figure 11 said 5% threshold. I think it should have been 10% according to the context of the manuscript.

Thresholds in Figure 11 (now Figure12) represent the thresholds that leave 90% empirical points IND above the curve for different range of duration. We corrected the figure caption and now they are the 10% thresholds.

The statement in lines 563-565 was wrong. There were points of the July 2008 event plotted on and below the threshold lines in Figure 11.

All the points that represent the mud-debris flows occurred in Valtellina on July 2008 are above both the single Sondrio Province curve and Sondrio+Brescia provinces. Two of the three points are plotted widely above the curves in Figure 11A (SO) and Figure 12 (SO, SO+BS) (now figure 12 and 13 respectively): the third point is plotted very near the curve. We realized that there were some problems in the previous figures. Consequently, Figure 11A and Figure 12 have been adjusted.

Figure 6: Fonts are too small to be legible. A sans-serif typeface such as Helvetica and Arial is preferred.

Figure 6 has been adjusted and a bigger font has been used.

Lines 25-27 (abstract): "Furthermore, we found that landslides occurred primarily … in a period from 5 to 15 hours in spring-autumn events." This is a wrong statement. According to Table 2, it should have been 0-5 hours.

According to Table 2, landslides occurred primarily in a period of 5 hours from the maximum, peak of rainfall intensity (43,6%) and, secondarily, from 5 to 15 hours (25,6%): we would have stress the landslides in spring-autumn events landslides occurred generally later than in summer period, emphasizing particularly when the gap between the maximum peak of rainfall intensity and the initiation of landslides is higher than 5 hours. However, the statement has been modified (in a period of 5 hours) as required.

Line 162: Did you mean "MAP"-normalized rainfall thresholds?

Bacchini and Zannoni (2003) defined a threshold for the possible initiation of debris flows in their study area (Dolomites, NW-Italy) in terms of mean intensity, duration and mean annual precipitation, i.e., they defined a MAP-normalized rainfall intensity – rainfall duration curve.

The sentence in the text has been clarified.  

Line 377: … temporal information "were" known.

“where” has been corrected in  “were”

Line 477: Mai is a typo.

We replaced “Mai” with “May” along the manuscript and the figure captions

Line 480: It should have been Figure 9B.

Figure number is correctly referenced (now figure 10B)

Line 493: It should have been Figure 9A. Figure number is correctly referenced (now figure 10A)

Reviewer 2 Report

This paper aimed to reconstruct rainfall thresholds for the occurrence of fast and shallow slope instabilities for two large areas of Central Italian Alps, namely in Lombardia region. The thresholds were reconstructed exploiting a hystorical 80-years database of rainfalls and times of occurrence of these instabilities, defining the thresholds by means of a typical empirical-statistical approach.

The work is very good, with a significant quality of data and important efforts of analysis and interpretations about the achieved results. The main limitation is the absence of a validation of the reconstructed models using an external database respect to that one used to obtain the thresholds.

Suggested revisions follow:

Europa has to be translated in English (Europe) throughout all the text Add more significant references about the effect of climate change in increasing or not shallow landslides occurrence. Moreover, add explanations on the reason why these effects could happen Why is Lombardy Alpine area significant for the entore Alpine area? Please, add several and more detailed explanations about this aspect In literature review about rainfall thresholds, it leaks completely references to thresholds reconstructed by means of physically-based approaches. Furthermore, few references to European and World contexts different to Italy are presented. Please, add also the most significant references about these aspects Explain better why you chose a buffer of 5 km to find the representative rain gauge for a particular shallow landslides/flows event Clarify how a rainfall event was distinguished from each other. Authors seem defined a dry period between two different events equal to 12 h. Is it correct? Why was this length chosen? In figure 7, it is required to add a legend which explain the meaning of the colours of the stars Mai in English is written as “May”. Please correct all along the manuscript The achieved thresholds should be validated, considering for example their predictive capability in idenitifying correctly triggering or not triggering conditions for a dataset different from the one used to reconstruct them. Maybe, validation dataset should cover the time span after 2008, as the Authors reconstructed the thresholds considering rainfall conditions of the years before that one. Then, you should also quantify the effectiveness of these thresholds, calculating statistical indexes such as Area Under ROC Curve, True Positive, False Positive, True Negative, False Negative What is the amount of cumulated rainfall required to trigger shallow instabilities in both the study areas, for particular durations typical of summer and spring-autumn events? Even if the triggering conditions are certainly different, I suggest to add, in the charts of Figure 13, also one or more thresholds defined for susceptible contexts of southern Italy (e.g. pyroclastic deposits of Campania region, Sicily region) and to comment the differences respect to the thresholds defined in this work

Author Response

Review Report Form 

Open Review

English language and style

( ) Extensive editing of English language and style required  
( ) Moderate English changes required  
( ) English language and style are fine/minor spell check required  
(x) I don't feel qualified to judge about the English language and style  

Yes

Can be improved

Must be improved

Not applicable

Does the introduction provide sufficient background and include all relevant references?

( )

(x)

( )

( )

Is the research design appropriate?

(x)

( )

( )

( )

Are the methods adequately described?

(x)

( )

( )

( )

Are the results clearly presented?

(x)

( )

( )

( )

Are the conclusions supported by the results?

(x)

( )

( )

( )

Comments and Suggestions for Authors

This paper aimed to reconstruct rainfall thresholds for the occurrence of fast and shallow slope instabilities for two large areas of Central Italian Alps, namely in Lombardia region. The thresholds were reconstructed exploiting a hystorical 80-years database of rainfalls and times of occurrence of these instabilities, defining the thresholds by means of a typical empirical-statistical approach.

The work is very good, with a significant quality of data and important efforts of analysis and interpretations about the achieved results. The main limitation is the absence of a validation of the reconstructed models using an external database respect to that one used to obtain the thresholds.

Suggested revisions follow:

Europa has to be translated in English (Europe) throughout all the text

Europe is now translated correctly in English throughout all the text

Add more significant references about the effect of climate change in increasing or not shallow landslides occurrence.

As required, we added more references about the correlation relation between climate change and its potential climate effects on the occurrence – or lack of occurrence – of landslides and the possible effects of climate changes in the frequency, extent or behaviour of landslides.

Moreover, add explanations on the reason why these effects could happen

We think that effect of climate changes on landslides occurrence is one of the most important and discussed issue nowadays. Analysis conducted in other areas of Northern Italy have shown us that there is a slight decrease in the cumulative annual rainfall and a marked decrease in the number or rainy days. This results in more violent rainfall in less days. Inevitably, these rains can create by superficial landslides and more frequent debris flows. However, this analysis would require further investigations and explanations but it lies outside the aim of our study, i.e.,  the definition of rainfall thresholds for the possible initiation of shallow landslides and mud-debris flows triggered in the eastern Alps.

Why is Lombardy Alpine area significant for the entore Alpine area? Please, add several and more detailed explanations about this aspect

The Lombard alpine sectors possess characteristics (geomorphological, lithological, hydrographical) very similar to those of the adjacent regions, which is why they can easily be taken as a significant example for the entire Alps. Anyway, we explained the importance of the Lombardy alpine sector within the entire alpine area, including some references. In particular, it represents one of the mountain area in the Alps and in Italy with the largest number of properties exposed to a high and very high landslide risk, where several shallow landslides and mud-debris flows occur every year causing considerable damage to structures and infrastructures and great losses in socio-economic terms.

In literature review about rainfall thresholds, it leaks completely references to thresholds reconstructed by means of physically-based approaches. Furthermore, few references to European and World contexts different to Italy are presented. Please, add also the most significant references about these aspects

As we affirmed in the section 2, we briefly described the most important works aimed to define the rainfall thresholds for the initiation of shallow landslides and mud-debris flows in mountain environment of different with particular regard to in the Italian Alps. Our aim was a literature review about rainfall thresholds in similar geographical environments to highlight similarities and differences between the various approaches adopted by authors to define the curves and to stress the importance of the historical research (80-year of landslides and rainfall data collected) for the determination of thresholds curves. It is emphasized that previous studies in Italy have NEVER considered such a long period of recorded data-set of rainfall.

Explain better why you chose a buffer of 5 km to find the representative rain gauge for a particular shallow landslides/flows event

We better explained the criteria adopted in the selection of the most representative rain gauges and why we choose a buffer of 5 km. A new figure has been added to highlight the considerable spatial variability in climate regime and how it may influence the rainfall measurements recorded by rain gauges located at similar geographical distance from a landslide.

Clarify how a rainfall event was distinguished from each other. Authors seem defined a dry period between two different events equal to 12 h. Is it correct? Why was this length chosen?

We clarified how we determine the rainfall duration in the study to avoid misinterpretation of the results, as suggested. By analysing the pluviogram recorded by each most representative rain gauge, we to determine the rainfall duration by measuring i) the period between the time when the rain started in the pluviogram and the time of the landslide occurrence and ii) a preceding period of minimum 12-hours with rainfall intensity lower than 1 mm/h. We considered both conditions to measure rainfall duration, except for some rainfall events extended over a several days period (more than 5-6 days), typically in spring and autumn months, for which we measured the rainfall duration using only the first condition. In these cases, we observed that the preceding 12-hours periods of very-low rainfall intensity (< 1 mm/h) were alternated by very heavy showers, which played a not negligible role in triggering landslides.

In figure 7, it is required to add a legend which explain the meaning of the colours of the stars

For each province we used different colours of the stars that represent landslides. Colours of the stars are now explained in the figure caption 8now Figure 8).

Mai in English is written as “May”. Please correct all along the manuscript

“Mai” has been modified in the correct form “May” along the manuscript and the figure captions

The achieved thresholds should be validated, considering for example their predictive capability in idenitifying correctly triggering or not triggering conditions for a dataset different from the one used to reconstruct them. Maybe, validation dataset should cover the time span after 2008, as the Authors reconstructed the thresholds considering rainfall conditions of the years before that one. Then, you should also quantify the effectiveness of these thresholds, calculating statistical indexes such as Area Under ROC Curve, True Positive, False Positive, True Negative, False Negative What is the amount of cumulated rainfall required to trigger shallow instabilities in both the study areas, for particular durations typical of summer and spring-autumn events?

We are thankful to the reviewer for his suggestion: we realized that the achieved thresholds should be validated. Several pluviometric events collected during the research in the historical annals were selected and set aside as "false negative" despite having exceeded the rain threshold.

However, the historical research had the task of finding a corresponding effect on the ground (before or after the rainfall data). In several cases nothing was found in the newspapers and in the reports of the municipalities. However, this does not necessarily mean that nothing had happened: perhaps after the shower a shallow landslide triggered on the slope accumulating material in a meadow without doing damage: in this case it is almost impossible to find the historical news.

Further data on rainfall-induced shallow landslides and debris flows occurred in the period after 2008 will be collected with the aim to validate the established thresholds and to quantify its effectiveness.

Even if the triggering conditions are certainly different, I suggest to add, in the charts of Figure 13, also one or more thresholds defined for susceptible contexts of southern Italy (e.g. pyroclastic deposits of Campania region, Sicily region) and to comment the differences respect to the thresholds defined in this work

To compare the rainfall thresholds established for the study area with other curves in literature, we adopted only similar IMAP-D thresholds, with a particular attention to similar alpine and mountain regions (see also the world-wide database of empirical rainfall thresholds for the possible occurrence of landslides compiled by CNR-IRPI available at:

http://wwwdb.gndci.cnr.it/php2/rainfall_thresholds/thresholds_type.php?typesigla=IMAP-D&lingua=it) Rainfall thresholds defined in literature for southern regions in Italy are usually ED thresholds, and we have no sufficient information about the MAP values in order to compare the different curves. Furthermore, we think that they have different geological, lithological, geographical and climatic settings and a comparison would be not suitable.

Round 2

Reviewer 1 Report

The authors have made a very good effort to address the issues raised in my previous review comments, and have revised their manuscript thoroughly. I am satisfied with their corrections and explanations. I think the paper should be accepted for publication now.